

# Suspended sediment concentrations in Alpine rivers: from annual regimes to sub-daily extreme events

Amber van Hamel[1, 2, 3], Peter Molnar[4], Joren Janzing[1, 2, 3], and Manuela Irene Brunner[1, 2, 3]

[1]WSL Institute for Snow and Avalanche Research SLF, Davos Dorf, Switzerland
[2]Institute for Atmospheric and Climate Science, ETH Zurich, Zurich, Switzerland
[3]Climate Change, Extremes and Natural Hazards in Alpine Regions Research Center CERC, Davos Dorf, Switzerland
[4]Institute of Environmental Engineering, ETH Zurich, Zurich, Switzerland

**Correspondence:** Amber van Hamel (amber.vanhamel@slf.ch)

**Abstract.** The occurrence of extreme suspended sediment concentrations (SSC) can have a detrimental impact on human infrastructure, water use, and the health of aquatic ecosystems. However, the majority of existing studies have focused on the SSC dynamics of individual catchments or single events, with the consequence that large-scale patterns of suspended sediment dynamics remain poorly understood. The objective of this study is to identify the principal factors that influence the spatial and temporal variability of suspended sediment concentration (SSC) and the occurrence of SSC extremes in the Alps. For our analyses, we use 10 years of observed sub-daily SSC data from 38 gauging stations in Switzerland and Austria. First, we examine spatial patterns in the annual median SSC regime, utilising hierarchical clustering based on the differences in magnitude, timing and shape of the annual SSC regime. The clusters are then reconstructed and explained by linking them to a large set of potential static hydro-climatic and catchment-related characteristics. This approach identified three distinct clusters of annual SSC regimes, for which the shape of the regime and the timing of the annual peak SSC are significantly influenced by catchment elevation, the start of the melt season, and the presence of glaciers. Second, we transition from the annual to the event scale at a sub-daily time step by identifying extreme events. We present a novel classification scheme that can be employed to categorise extreme SSC events and differentiate between nine event types based on their dominant transport processes. We examine the spatial and temporal distribution of these nine event types across the Alps, the severity of the events, and the effect of antecedent conditions, such as snow cover, soil moisture, and catchment memory. The analysis of 2,398 extreme SSC events across all catchments indicates that rainfall is the primary driver of SSC extremes, responsible for 80 % of all events. Nevertheless, in high-elevation and partially glaciated catchments, up to 40 % of the events are attributed to snow and glacial melt, underscoring the disproportionate impact of meltwater on sediment concentrations in Alpine rivers. The combination of high-intensity rainfall and glacial melt events resulted in the highest SSC and second highest area-specific suspended sediment yields (sSSY) on average among all event types. A notable proportion of the extreme events (24 % of the total) resulted in peak SSC values exceeding 5 g L$^{-1}$, highlighting their potential to significantly harm aquatic species and riverine ecosystems. Our findings underscore the importance and impact of extreme SSC events on water quality and sediment transport in Alpine river systems.



## 1 Introduction

The total sediment transported by rivers generally consists of suspended sediment and bedload. The fraction of suspended sediment in the total sediment load can vary greatly as a function of sediment supply, lithology, and catchment size, but on an annual timescale the flux of fine sediment in suspension is often the dominant sediment transport process (Turowski et al., 2010). This study focuses on the suspended sediment concentration (SSC) of rivers in the European Alps. Although suspended sediment is a natural component of rivers, extreme concentrations can have significant impacts on hydropower generation and

reservoir sedimentation (Panagos et al., 2024), flood impacts (Nones and Guo, 2023; Vázquez-Tarrío et al., 2024), and water quality (Coffey et al., 2019), including the costs of water treatment and the transport of nutrients and contaminants (Brighenti et al., 2019; Steingruber et al., 2021; Zaharescu et al., 2016). In addition, extreme concentrations can have adverse effects on riverine ecosystems and aquatic species as they reduce water quality, cause a decline in transparency and sunlight penetration, and result in the clogging of the gills of fish and other aquatic organisms, which can ultimately have lethal impacts (Bilotta

et al., 2012; Kemp et al., 2011; Newcombe and Macdonald, 1991). Given that the majority of suspended sediment is transported during a few extreme events (Blöthe and Hoffmann, 2022; Schmidt et al., 2022), it is essential to gain a deeper understanding of the spatial and temporal dynamics of suspended sediment concentration and its extreme conditions.

In mountain environments, there are three hydro-climatic processes that are most important for sediment transport to the river, namely, erosive rainfall, snowmelt and glacial melt (Costa et al., 2018a). Together, these processes are responsible for

the detachment and erosion along hillslopes and for connecting the sediments to the river network by overland flow. They can also trigger mass wasting events such as debris flows and landslides, which can mobilise even greater amounts of sediment and potentially lead to very high concentrations of suspended sediment in the river (Battista et al., 2022). In addition, the combination of rainfall, snow- and glacial melt, and groundwater inflow, controls the river discharge, which determines a river's transport capacity. High flows keep sediment in suspension, allowing it to be transported downstream, and can lead to

additional sediment inputs through channel and bank erosion (Park and Hunt, 2017) and the re-suspension of particles that have been stored in the river bed (Deng et al., 2024).

Climate plays an important role in controlling sediment erosion and the main transport processes. Changes in rainfall, melt-water, snow cover, permafrost and glacier retreat will likely have an impact on erosion along hillslopes, sediment availability, sediment connectivity with the stream, and the amount of energy available for sediment transport (Hirschberg et al., 2021;

Costa et al., 2018b; Mishra et al., 2020; Mancini et al., 2024). To assess future changes in SSC, a better understanding of the relationship between these climatic processes and the SSC is needed. This requires a comprehensive understanding of the current spatial and temporal variations in the general SSC behaviour, as well as the specific characteristics of extreme SSC events. Nevertheless, there is still limited quantitative understanding of sediment transport in alpine rivers, largely due to the complex, non-linear and highly stochastic nature of suspended sediment transport. This complexity is a consequence of the many poten-

tial hydro-climatic drivers of sediment transport and their interplay, the numerous processes that regulate sediment production and availability, and the continuous changes in sediment connectivity. Furthermore, the suspended sediment concentration in rivers is significantly influenced by unpredictable mass wasting, such as landslides (Battista et al., 2022).



To date, large-scale, multi-catchment analyses investigating the seasonal and sub-daily dynamics of SSC are scarce. A considerable number of studies have attempted to predict the seasonal dynamics of sediment transport in different catchments based on catchment characteristics and hydro-climatic factors (Doomen et al., 2008; Mano et al., 2009; Schmidt et al., 2022; Costa et al., 2017; Sutari et al., 2020). However, these studies have primarily focused on individual catchments or specific events, limiting generalizability and our understanding of these processes at larger scales. Those studies that did compare sediment transport across different catchments, either relied on data sets with limited temporal resolution, i.e. monthly and annual observations (Hinderer et al., 2013; Li et al., 2020), or focused only on sediment yield without considering sediment concentration (Haddadchi and Hicks, 2020). In order to gain insight into the relative severity and impact of suspended sediment events, it is necessary to consider both the suspended sediment yield (SSY) and the SSC. The former describes the actual amount of sediments that is transported by the river, affects river morphology, and is an indicator of soil loss through water erosion. The latter is a measure of turbidity and activation of sediment sources and therefore affects water quality and river ecology.

Event-scale studies have also received considerable attention in the literature, and the majority of these studies make use of sediment-discharge relationships and hysteresis loops to estimate sediment sources, suspended sediment delivery mechanisms, and landscape connectivity. Although these methods are widely used, they are known to have several limitations. The SSC-discharge relationship (also called the sediment rating curve) is purely empirical and is not able to account for (seasonal) variations in sediment availability (e.g. stocking, depletion and activation of different sediment sources) (Doomen et al., 2008; Horowitz, 2003; Zhang et al., 2021; Costa et al., 2018b). Most studies still use baseflow separation techniques to extract SSC events (Shin et al., 2023; Haddadchi and Hicks, 2021; Blöthe and Hoffmann, 2022; Skålevåg et al., 2024), approaches that often neglect events with high SSC but low discharge. This is a problem from a water quality perspective, as such events are similarly relevant as events associated with high discharge. Furthermore, the use of sediment-discharge hysteresis loops, which show the relationship between discharge and SSC during an event, has been popular for classifying events and estimating sediment sources (Walling, 1977; Millares and Moñino, 2020). Although the hysteresis method is commonly applied, its results are highly location dependent and the interpretation of hysteresis patterns is challenging due to the influence of feedback mechanisms, interactions between multiple drivers, and the uncertainty regarding the location of sediment sources within the catchment (Skålevåg et al., 2024; Misset et al., 2019). Based on an event classification by Skålevåg et al. (2024), hysteresis explains only about one tenth of the variability of sediment-discharge events. The above limitations calls for a novel approach for event classification that does not rely on rating curves and hysteresis patterns. Furthermore, there is a clear need for a more comprehensive understanding of annual and sub-daily patterns in SSC, as well as the occurrence of SSC extremes, across a large domain. Such large-sample analyses could improve our understanding of sediment dynamics not just at the local scale but enable generalizations across multiple catchments.

To address these research gaps, this study aims (i) to quantify spatial and temporal differences in the annual SSC regime among catchments and explain these patterns based on static catchment characteristics, (ii) to design a classification scheme to distinguish between different extreme SSC event types, and (iii) to explain the spatial and temporal differences between extreme SSC event types based on event characteristics, time-varying characteristics, and antecedent conditions. In this study,



we assume that time-varying hydro-climatic processes and catchment characteristics, such as changes in precipitation, snow
cover and soil moisture, are responsible for the local variation and the inter-event variability of SSC, while static characteristics
such as geology, altitude and mean annual temperature may be important in explaining the external spatial variation in SSC
dynamics between catchments.

The novelty of this study is fourfold: (1) We examine key indicators of the annual SSC regime and extreme SSC events
for 38 catchments in the Alpine region in order to assess the spatio-temporal variability of SSC over a large region; (2) We
include hydro-climatic forcing (precipitation, snowmelt and glacial melt) prominently in our analyses, event selection and
event classification, as an alternative to the more traditional discharge-based approach; (3) We link annual patterns to sub-daily
extremes; and (4) we introduce an event classification scheme that is independent of the SSC-discharge relationship and can
be applied to any type of catchment, making it applicable to other regions.

## 2   Methodology

To study spatial variations in seasonal and event-based sediment patterns, we performed two main analyses: (i) we quantified
spatial and temporal variations in the annual SSC regime and explained the patterns based on static catchment characteristics,
and (ii) we explained the spatial and temporal variations in SSC events based on event characteristics, time-varying hydro-
meteorological and catchment characteristics, and antecedent conditions. In order to analyse SSC events, we introduced a new
classification scheme to distinguish between different event types.

### 2.1   Data sets

### 2.1.1   Suspended sediment concentration (SSC) data

For this study, we used data from 38 catchments in the Swiss and Austrian Alps for which turbidity-based suspended sediment
concentration data (SSC) were available. These catchments cover different elevations and sizes, with the mean catchment
elevation varying from 321 to 2858 m.a.s.l. and catchment areas varying between 22 to 6297 km$^2$.

For the Austrian stations, we made use of quality-checked 15-minute SSC data for the period 2009–2021 provided by
the Austrian Hydrographic Service (Habersack et al., 2017). These data were created by merging manually conducted bi-
weekly SSC samples with 15-minute turbidity measurements that are automatically obtained by optical infrared turbidity
sensors (Solitax sensors by Hach, concentrations in mg L$^{-1}$). Combining SSC samples with turbidity data to obtain a high-
frequency SSC data set is a commonly applied and accepted method, because of the strong relationship between turbidity and
SSC (Gippel, 1995; Grayson et al., 1996). For more details on the methods used by the Austrian Hydrographic Service, see
Habersack et al. (2017).

For the Swiss stations, bi-weekly SSC samples and 10-minute turbidity data were provided by the Federal Office for the
Environment (FOEN) for the period 2014–2023. Unlike the Austrian data, the data had not been quality checked and we had
to reconstruct the turbidity-based SSC data ourselves. Due to the gradual replacement of older sensors (which were incapable





of correctly detecting high turbidity levels >1000 NTU) with newer and more accurate sensors, the data cleaning resulted
in the removal of turbidity levels above 1000 nephelometric turbidity units (NTU) for the period before 2019. Additionally,
implausible outliers were removed when the SSC samples were more than 100 times larger than the turbidity-derived values, or
vice versa, which resulted in the removal of, on average, three to four values per station. Two further implausible outliers were
removed at the stations of Porte du Scex (Rhône, id 2009) and Bellinzona (Ticino, id 2020), where turbidity values exceeded
3000 and 5000 NTU, respectively.

The relationship between SSC and turbidity can be approximated by a linear or non-linear model (Gippel, 1995). A linear
model assumes a constant relationship between turbidity and SSC, indicating a constant shape and density of the suspended
sediments over time, while a non-linear model is preferred in situations where particle size varies with concentration and
the relationship between turbidity and SSC is no longer linear (Gippel, 1995). A slightly modified approach applies a log-
transformation to the turbidity and SSC data before fitting such a non-linear model (Costa et al., 2018a). One downside of this
approach is that back-transformation from the logarithmic to the actual scale requires an additional bias correction function
(Duan, 1983), to correct for the larger positive residuals that are a result of the initial log-transformation. In order to find the
best SSC-turbidity relationship for our Swiss stations, we fitted a linear model, a non-linear model and a non-linear model after
logarithmic transformation for each station separately, considering simultaneous measurements of turbidity and SSC (with a
maximum time lag of 10 min). For all stations, the model performance was the highest for the non-linear model with the form
(Supporting Information Table S1 and Figure S1):

$$\text{SSC} = a * \text{Turb}^b, \tag{1}$$

where $Turb$ stands for the turbidity in NTU, $a$ is a characteristic coefficient, and $b$ reflects the effects of systematic variations
in particle composition with concentration (Gippel, 1995). The calibrated parameters ($a$ and $b$) had to be estimated for each
catchment individually (Supporting Information Table S2). We computed 10-minute SSC time series for each Swiss catchment
by applying these catchment-specific non-linear models to the 10-minute turbidity measurements.

    Finally, we averaged the Swiss (10-minute) and Austrian (15-minute) SSC data to an hourly time step, as most of the hydro-
climatic data are not available on a finer time scale. Another benefit of this hourly resolution is that it reduced the influence of
single outliers. While we used hourly data for the analysis of extreme SSC events, we further aggregated the data to daily SSC
for the analysis of the annual SSC regime. The selected 38 stations all have a time series length of 10 to 12 years within the
period 2009–2023.

### 2.1.2   Hydro-climatic data

In addition to the SSC data, we also obtained some other hydro-climatic datasets from different sources. The Austrian Hydro-
graphic Service and the Swiss FOEN provided observational discharge measurements for the 38 stations, which we use at an
hourly time step. Hourly precipitation data were available based on a geostatistical combination of rain gauge measurements
and radar estimates for Switzerland (CombiPrecip provided by MeteoSwiss at a 1 km$^2$ resolution and available for 2005–2024,



(Gabella et al., 2017)) and for Austria (INCA provided by the Central Institute for Meteorology and Geodynamics (ZAMG) at a 1 km² resolution and available for 2007–2018, (Haiden et al., 2011)).

Daily snowmelt and ice melt at a 30 arcsec (approx. 1 km at the equator) grid scale were derived from simulations generated by the gridded global hydrological model PCR-GLOBWB 2.0 (Sutanudjaja et al., 2018; Hoch et al., 2023) for the period

1990-2019. This model was adapted and evaluated for the Alps by (Janzing et al., 2024) and contains an updated snowmelt routine (with an expanded temperature index model and regional calibration) and a new glacier routine. In addition to snow- and glacial melt, we calculated daily percentages of snow-free and snow-covered areas for each catchment from snow water equivalent (SWE) data generated by the PCR-GLOBWB 2.0 model, where a grid cell is assumed to be free from snow when SWE < 0.1 mm.

Another variable of interest is soil moisture, as partly saturated soils result in more surface runoff and therefore erosion and the input of fine sediment. Very saturated soils could be less stable and more prone to landslides (Godt et al., 2009; Gariano and Guzzetti, 2016). Daily liquid volumetric soil moisture is obtained from the gridded Copernicus European Regional ReAnalysis Land (CERRA-Land) dataset (Verrelle et al., 2022), which has a high temporal and spatial resolution of 3 h and 5.5 km, respectively, for the period 1984-2021.

## 2.2  Static and time-varying catchment characteristics

Characteristics that are relevant for suspended sediment dynamics in rivers can be divided in two groups: (i) static characteristics (such as geology, elevation, and mean temperature) that explain the external variation in the SSC regimes between catchments and (ii) time-varying characteristics (such as precipitation and snowmelt) that explain the inter-event variability in SSC within one catchment.

### 2.2.1  Static characteristics

Static characteristics (stable in time) include catchment characteristics, such as elevation, but also hydro-meteorological attributes, such as mean daily temperature. These static characteristics can either control or affect the sediment transport processes or the sediment availability in a catchment. Characteristics related to sediment transport are for example the slope, runoff-ratio, mean daily discharge, and mean daily precipitation. Characteristics related to sediment availability are for exam-

ple the land cover types and geology classes that are present in the catchment. Figure S2 and S3 in the Supporting Information provide an overview of all static characteristics that we considered in this study. Some of these characteristics were provided by the large-sample hydrological data sets Camels-CH (Höge et al., 2023) and LamaH-CE (Klingler et al., 2021) and others have been added based on our own calculations. For example, we grouped the 13 geology classes, as defined by GLiM (Hartmann and Moosdorf, 2012), in three groups based on their erodibility index (Alps in Table 1 of Moosdorf et al. (2018)): a low,

median, and high erodible geology class. In addition, we calculated the elongation ratio, stream density and meandering index in QGIS based on the DEM (Copernicus, 2013) and catchment delineations, to capture information on fluvial geomorphology (for detailed explanation of these characteristics, see Supporting Information Figure S2 and S3). Finally, we calculated the per-





centage of the catchment area that was located upstream of big lakes and (hydropower) reservoirs ($> 1$ km$^2$) as these regions might contribute less to the SSC in rivers when sediments get trapped or settle in those larger water bodies.

### 2.2.2 Time-varying characteristics

Time-varying characteristics capture inter-event variability and can also be divided into those related to sediment transport and those related to sediment availability. An important catchment characteristic that changes per SSC event and affects the sediment availability is the actively contributing drainage area. The active drainage area, that can drain water and mobilize sediments, is small when a large part of the catchment is covered by snow. In contrast, snow-free areas are potentially erodible under the assumption that the ground is largely unfrozen and thus susceptible to erosion (for example by precipitation and overland flow). Other time-varying catchment characteristics of interest are changes in soil moisture and changes in sediment availability (depletion and replenishment of sediment sources). Time-varying hydro-climatic characteristics that relate to sediment transport processes are hourly and daily (erosive) precipitation, snowmelt, ice melt, and river discharge. Figure S4 in the Supporting Information provides an overview of all time-varying characteristics that we considered in this study.

## 2.3 Spatial and temporal differences in the annual sediment regime

To explain spatial and temporal variations in the annual SSC regime among catchments, we made use of hierarchical clustering. The methodology comprised several steps which are discussed in more detail below: (i) defining magnitude-shape-timing (MST) indicators that characterise variations in the annual median SSC regime, (ii) clustering the catchments based on these MST indicators, (iii) identifying the most relevant static characteristics that explain the variation in annual SSC regimes.

### 2.3.1 Indicators used to describe the annual SSC regime

The median annual SSC regime is defined as the 50th percentile of mean daily SSC, smoothed over a 30-days time window to limit the sensitivity of the annual SSC regime to single values. Next, we selected five indicators that are able to capture the variations in magnitude, shape and timing of the annual SSC regime, referred to as MST-indicators (Table 1 and Figure 1a). The magnitude is represented by the median annual SSC per catchment (*SSC_mean*). Shape indicators include the number of peaks (*Peak_numb*) and the relative magnitude difference between the highest and second highest peak (*Peak_magdiff*). Timing indicators include the day of the year (*Peak1_doy*) on which the highest peak occurs and the time difference in days between the highest and second highest peak (*Peak_timediff*). The main peaks were identified automatically with the *scipy.signal.find_peaks* package (Virtanen et al., 2020) based on three criteria: peaks only occur above the 75th percentile of the annual SSC regime; the minimal horizontal distance between peaks is 30 days (one month); and the peak prominence, which "measures how much a peak stands out from the surrounding baseline of the signal and is defined as the vertical distance between the peak and its lowest contour line" (Virtanen et al., 2020). In our case, the peak prominence is catchment specific and defined as 5 % of the SSC range ($0.05 * (SSC\_max - SSC\_min)$). These parameters for peak identification were selected after visual inspection, with at least one peak identified per SSC regime and without selecting too many additional peaks that did not contribute to




the overall shape. The number of identified peaks varied from one to three peaks per annual SSC regime. For the catchments

with multiple peaks, we also calculated the time lag and relative magnitude difference between the highest and second highest

peaks. When the time difference (*Peak_timediff*) has a positive value, the highest peak happens earlier in the year than the

second highest peak, and vice versa.

**Table 1.** Overview of the five selected MST-indicators that reflect the magnitude, timing and shape of the median annual SSC regime per

catchment.

|  | Indicator | Description |
|---|---|---|
| **Magnitude** | SSC_mean | Mean daily SSC based on the annual 50th percentile SSC regime. |
| **Shape** | Peak_numb | The number of main peaks (values between 1 and 3). |
|  | Peak_magdiff | The relative magnitude difference between the highest peak and the second |
|  |  | highest peak (note: only if Peak_numb >1), by dividing the absolute difference |
|  |  | derived by the range of the SSC regime: (Peak1_ssc - Peak2_ssc) / (SSC_max - SSC_min) |
| **Timing** | Peak1_doy | The day of the year (doy) with the highest SSC peak. |
|  | Peak_timediff | The time difference in days between the highest peak and the second highest |
|  |  | peak (note: only if Peak_numb >1). |

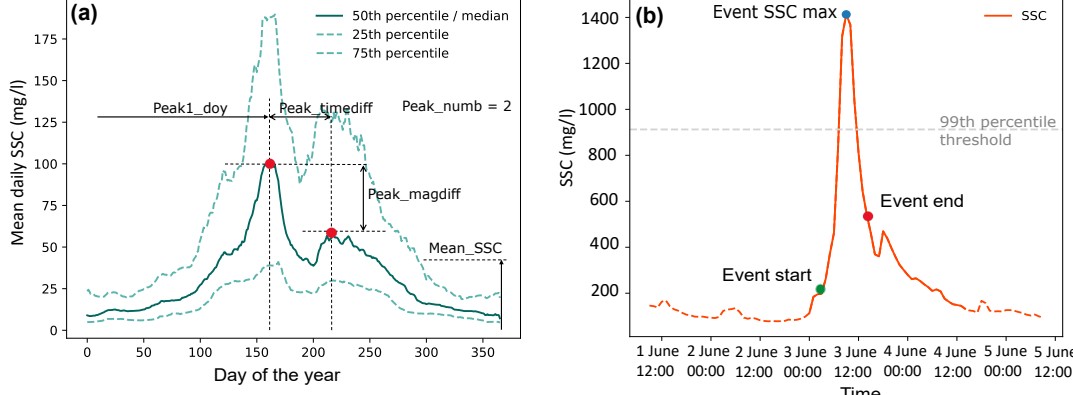

**Figure 1.** (a) Visual overview of the five selected MST-indicators (explained in Table 1) in relation to the median annual SSC regime of one

example catchment. (b) SSC event (with event start and end), where the SSC max is exceeding the 99th percentile threshold.

### 2.3.2 Hierarchical clustering based on SSC regime indicators

We applied hierarchical clustering to cluster the catchments based on their MST-indicators. Clustering started by computing the

Euclidean distance matrix, using the matrix of $n = 38*5$ standardized MST indices (python package *sklearn.preprocessing.StandardScaler*)





Next, we used a hierarchical clustering algorithm (python package *sklearn.cluster.AgglomerativeClustering*), that allows for non-elliptical clusters, and the Ward variance minimization algorithm, which minimizes the total within-cluster variance (Ward Jr., 1963), to identify clusters of similar SSC regimes. Finally, we optimized the number of clusters ($k$) by applying the silhouette score and the elbow method, and validated the clustering based on a visual inspection of the dendrogram, keep-
ing in mind the interpretability of the clusters. A comparison with clusters derived by $k$-means clustering showed that the final clusters identified are relatively stable independent of the choice of the clustering technique.

### 2.3.3 Interpretation of variation in SSC regimes based on static catchment characteristics

For the interpretation of the three SSC regime clusters, we selected a number of static catchment characteristics and hydrom-eteorological attributes (Section 2.2.1), that may be able to explain the spatial and temporal variations in annual SSC regimes
in mountain rivers. To better compare these characteristics among the different catchments, we used area-independent val-ues, such as percentages (e.g. % forest cover) and specific discharge (in mm d$^{-1}$). In addition, we normalized all catchment- and hydro-meteorological attributes to the range 0-1 across all catchments. For each cluster, we identified the most important static characteristics that are able to explain the spatial variation between different annual SSC regimes. The most important characteristics are those that clearly differ for the selected clusters and thus support the clustering. Finally, we also compared
the annual regimes of the main transport processes (precipitation, glacial melt, snowmelt, and discharge) with the annual SSC regimes.

## 2.4 Spatial and temporal variations in extreme SSC events

To identify different types of extreme SSC events and explain the spatial and temporal occurrence of these events, we performed several analysis steps which are discussed in more detail below: (i) detecting extreme SSC events (ii) designing a classification
scheme to distinguish between event types, (iii) explaining the spatial and temporal differences between SSC event types based on time-varying characteristics, event characteristics, and antecedent conditions.

### 2.4.1 Detection of extreme SSC events

Extreme SSC events, i.e. episodic peaks of suspended sediment concentration, can be defined in different ways. While there is no commonly used definition for SSC extreme events, the peak-over-threshold method is widely used to extract events
(Skålevåg et al., 2024; Hamshaw et al., 2018; Haddadchi and Hicks, 2021; Blöthe and Hoffmann, 2022). In this study, we defined SSC extreme events as events for which the peak value exceeds a locally defined 99th percentile threshold (Figure 1b). We identified the start of the event based on a rapid increase in the slope of SSC prior to the SSC peak (the increase in slope [$\Delta$ mg L$^{-1}$] is larger than the difference between the 50th and 75th percentile of SSC for that catchment [mg L$^{-1}$]). The end of the event is defined as the time when the SSC drops below 0.4 times the SSC peak value. We also tested and compared other
definitions, where the end of the event was equal to crossing the 90th percentile or was determined based on the decrease in slope. However, a sensitivity analysis showed that the choice of the exact definition had a limited effect on the total number of




events selected (Supporting Information Figure S5 and Table S3). When the end of one and the start of the next event are less than 12 hours apart, these events were merged into one event, because we assumed that these events share the same drivers. We have deliberately chosen to make the selection of SSC events independent of discharge because such an approach also enables the selection of extreme SSC events during low flow periods. Events with missing SSC or hydro-climatic data are removed. Since daily snowmelt and ice melt data was only available until the end of 2019, this means that we did not include extreme events that occurred after this date. Our approach led to the extraction of 2398 SSC extreme events across the entire study domain (on average 73 events per catchment).

### 2.4.2 Classification of extreme events based on their dominant transport process(es)

To distinguish between different types of extreme events, we developed a classification scheme that identifies the most important driver(s) of each event. The three hydro-climatic processes that are most important for sediment transport to the river are erosive rainfall, snowmelt and glacial melt (Costa et al., 2018a). We assumed that each event is caused by either one or two out of these main transport processes. Inspired by the event-based classification of flood generation processes by Stein et al. (2020), we developed a classification scheme for extreme SSC events that assigns each event to one out of nine event types (Figure 2). The most dominant transport process(es) per event is (are) determined based on the total rainfall, snowmelt and glacial melt during the 24 hours prior to the SSC peak of the event. Costa et al. (2018a) found that a 1-day time lag is the most relevant time lag for predicting SSC for the Rhône catchment. We tested this time lag on a selection of events and found that this short time lag aligns well with the fast response time of our catchments. Furthermore, we distinguished between high- and low-intensity precipitation as two different driving processes, because soil erosion by runoff depends on rainfall intensity (Parsons and Stone, 2006; Mohamadi and Kavian, 2015). Precipitation is considered high intensity if the maximum hourly precipitation during the 24-hour time window exceeds the catchment-specific 99th percentile threshold of hourly precipitation. All other precipitation values are considered low intensity. According to our definition, an event had two dominant transport processes (e.g. precipitation and snowmelt driven) if both precipitation and snowmelt contributed more than 1/3 of the total water input. The threshold of 1/3 was chosen to allow for a maximum of two dominant processes per event. A lower threshold could result in the selection of all three drivers for an event, but information about which of the drivers is the most dominant would be lost. A sensitivity analysis showed that higher thresholds (of up to 1/2) result in a larger number of events with only one dominant transport process. Because we were also interested in events with compounding drivers, we used the 1/3 threshold.

### 2.4.3 Event characteristics and antecedent conditions analysed per event type

To learn more about the differences and similarities between the nine event types, we also looked at the spatial and temporal distribution of the event types across the Alps, the effect of antecedent conditions, and whether events belonging to the same event type share the same event characteristics. Event characteristics include event duration, mean and maximum event SSC, specific suspended sediment yield (sSSY) [t km$^{-2}$], and event complexity, defined as the number of peaks in the SSC between the start and end of the event (Figure 1b). In addition, we monitored important time-varying antecedent conditions during the



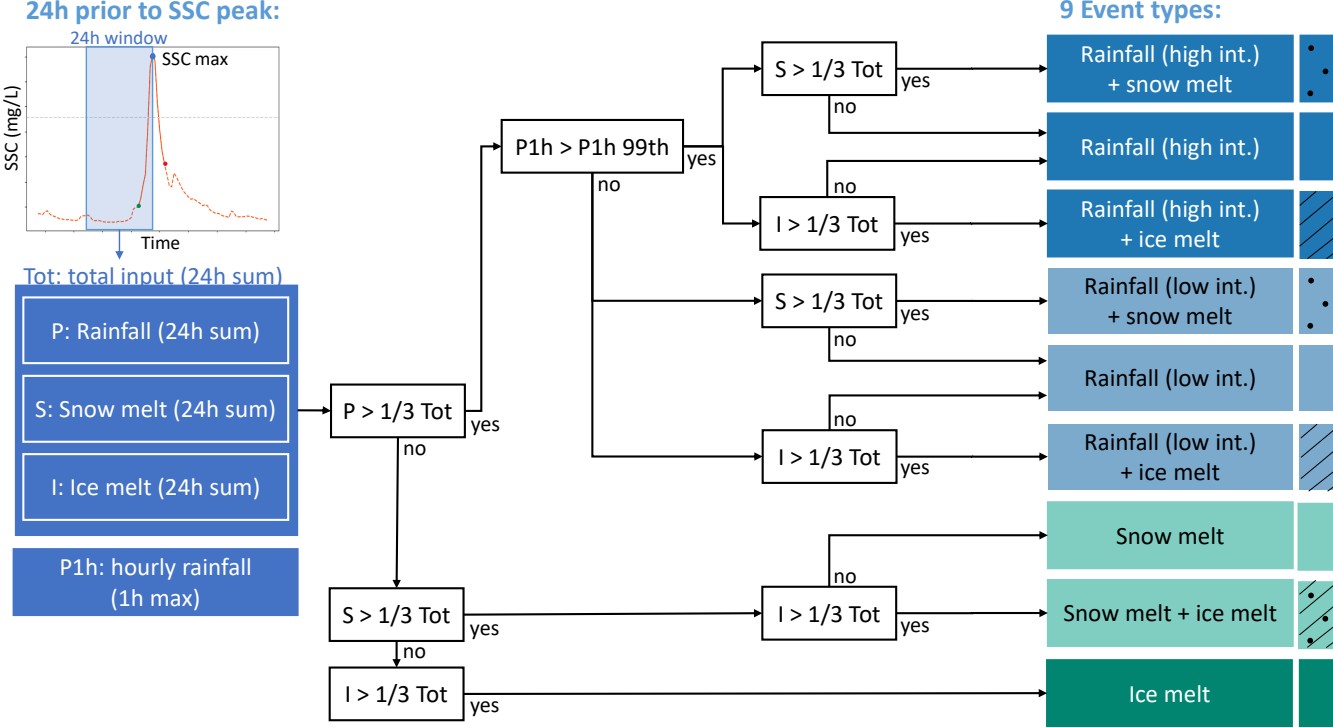

**Figure 2.** Decision tree used to categorise extreme SSC events into one out of nine predefined event types. The most dominant transport process(es) is (are) determined based on the fraction of total rainfall (P), snowmelt (S) and ice melt (I) (also referred to as glacial melt) compared to the total sum of P, S and I (Tot) during the 24 hours prior to the SSC peak of the event. Precipitation is classified as high intensity (high int.) if the maximum hourly precipitation (P1h) during the 24-hour period exceeds the catchment-specific 99th percentile threshold of hourly precipitation. Otherwise it is classified as low intensity (low int.).

2 days prior to the SSC event, such as the mean snow cover and the soil moisture in the catchment, which control sediment availability. Finally, we investigated the role of long-term catchment memory on extreme event development. As a proxy for sediment memory, we compared the long-term cumulative sSSY prior to each event with the catchment specific mean annual cumulative sSSY curve. For each day of the year, we determined whether the cumulative sSSY was above or below the mean annual sSSY curve. If the cumulative sSSY was below the mean (=negative deviation), less sediment than usual has been transported during that season and therefore more sediment may be available for mobilization during upcoming events. If the cumulative sSSY was above average (=positive deviation), more sediment than usual had already been transported by previous events, which may have led to sediment depletion in the catchment and limited sediment availability for the next events. To make this index comparable among catchments, we divided the deviation of the long-term cumulative sSSY from the catchment specific mean annual sSSY curve by the catchment annual mean cumulative sSSY.



**Table 2.** Statistical description of the variation in the annual SSC regime (median and maximum) and cumulative annual sSSY among the 38 catchments.

|  | Averaged over all 38 catchments | | |
|---|---|---|---|
|  | Median SSC-regime (mg L$^{-1}$) | Max SSC-regime (mg L$^{-1}$) | Annual sSSY (t km$^{-2}$ yr$^{-1}$) |
| Min | 2 | 4 | 23 |
| Median | 17 | 88 | 227 |
| Mean | 27 | 164 | 277 |
| Max | 229 | 1082 | 1415 |

## 3 Results

### 3.1 Large variation in median SSC among catchments

We observe a large variation in median SSC among the different catchments, with some catchments showing a median SSC which is a hundred times greater than the one of other catchments, even though all catchments are located in the same mountain range (Table 2). The median SSC varies from 2 mg L$^{-1}$ to 229 mg L$^{-1}$ between the 38 catchments, with a mean SSC value of 27 mg L$^{-1}$ on average. The mean annual sSSY varies from 23 t km$^{-2}$ yr$^{-1}$ to 1415 t km$^{-2}$ yr$^{-1}$, again with a difference of 2 orders of magnitude. The mean annual sSSY averaged over all 38 catchments is 277 t km$^{-2}$ yr$^{-1}$. The catchment with the largest median SSC and sSSY is the Vent catchment located in the Ötztal (Austria). This catchment has the highest mean elevation (2889 m.a.s.l.), a glacier cover of 30 %, and is know for its high sediment input (**?**Skålevåg et al., 2024). The catchment with the lowest median SSC is located at the foots of the northeastern Austrian Alps with a mean catchment elevation of 938 m.a.s.l.. The large spatial variability allows us to investigate the causes of variation between catchments. To do this, we group the catchments into different clusters and analyse these clusters separately.

### 3.2 Three types of annual SSC regimes

Our hierarchical clustering approach identified three main types of annual SSC regimes, based on the magnitude, the shape and the timing of the peak(s) (Figure 3 a-c): SSC regimes in Cluster 1 ($n =$12) have two or multiple peaks, where the highest peak occurs at the end of spring and earlier in the year than the second highest peak; SSC regimes in Cluster 2 ($n =$7) generally have one main peak, which occurs in mid-summer; and SSC regimes in Cluster 3 ($n =$19) show an opposite shape compared to Cluster 1, with the highest peak occurring later in the year than the second highest peak. The number of peaks and the timing of the first peak are the most important indicators for dividing the annual SSC regimes into these clusters, while the magnitude of the SSC regime (median SSC) is considered the least influential for clustering the catchments.

The catchments belonging to the three different SSC clusters are characterized by substantially different catchment characteristics (for an important subset see Figure 4a-h and for the full overview see Supporting Information Figure S6). The catchments that belong to Cluster 2 can be described as high-elevation, small mountain catchments because they are charac-




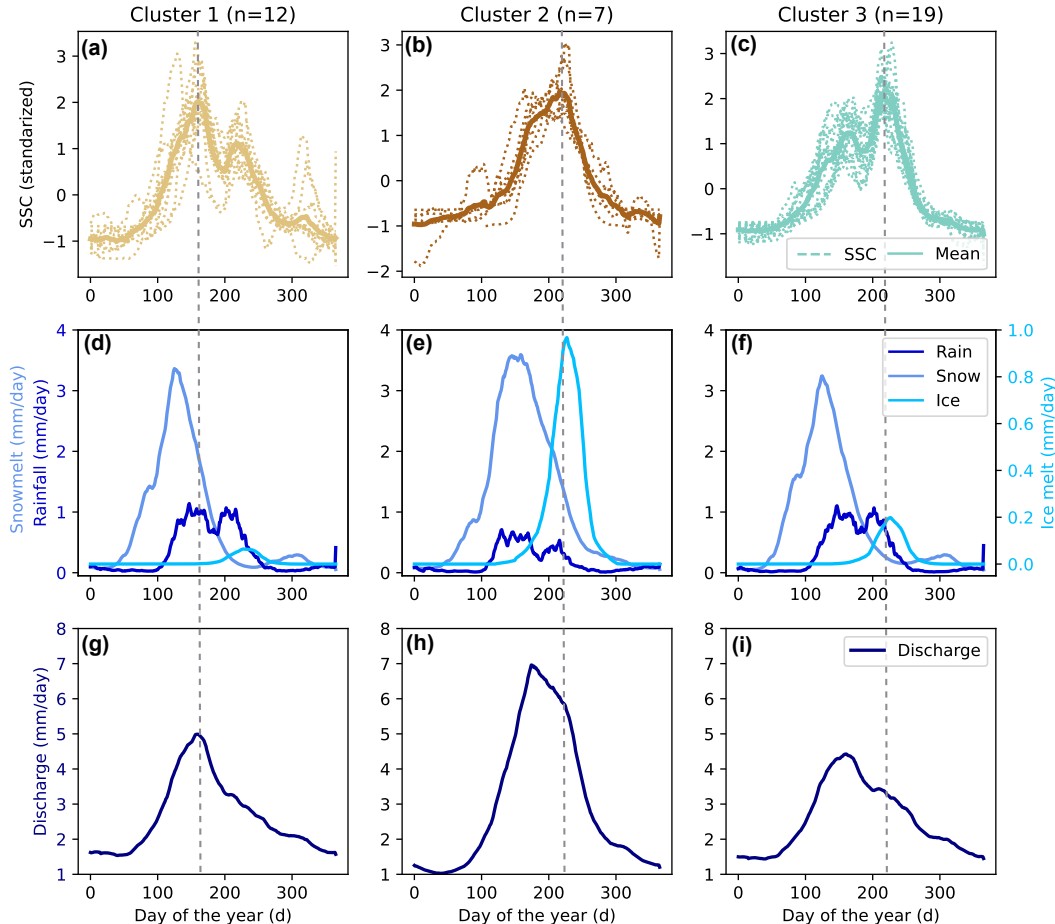

**Figure 3.** The median SSC regimes (standardized by removing the mean and scaling to unit variance) can be grouped into three different clusters: (a) Cluster 1 ($n=12$), (b) Cluster 2 ($n=7$), and (c) Cluster 3 ($n=19$). The dashed lines show the regimes for the $n$ individual catchments that belong to that cluster and the bold line is the mean of all catchments per cluster. Panels d, e, and f show the mean annual regimes of precipitation, snowmelt, and glacial melt averaged in mm/day over all catchments within a cluster. The mean annual discharge regime over all catchments within a cluster is given in panels g, h, and i. The grey vertical dashed lines spanning panels indicate when peak SSC occurs.

terised by high mean elevation and slope, a high fraction of glacier coverage, high mean daily snowmelt and glacial melt, and a high fraction of sand (dark brown boxplots in Figure 4a-h). The effect of reservoirs is limited and none of the catchments are located downstream of big lakes (Supporting Information Figure S6). The catchments belonging to Clusters 1 and 3 have quite similar static catchment characteristics (beige and turquoise boxplots in Figure 4a-h). Compared to those in Cluster 2, they have a larger area, a lower mean elevation and a higher fraction of clay and silt. The catchments in Cluster 1 have in general a smaller fraction of precipitation that falls as snow, and the day of the year on which the catchment is completely snow-free happens earlier in the year than for the catchments in Cluster 3. The fraction of highly erodible geology (such as unconsoli-






dated sediments) is higher in catchments belonging to Cluster 1, while the catchments in Cluster 3 have a larger fraction of low erodible geology (e.g., plutonic rocks, carbonate sedimentary rocks, acid volcanic rocks). Finally, a larger fraction of the catchments in Cluster 1 than Cluster 3 are located upstream of lakes and reservoirs (Supporting Information Figure S6).

The subtle differences between catchments in Clusters 1 and 3 become more pronounced when we compare the annual SSC regimes with the mean annual regimes of precipitation, snowmelt and glacial melt per cluster (Figure 3d-f). Catchments with a

single-peak SSC regime (Cluster 2) are characterised by late snowmelt (as high mean elevation results in a late melting season), immediately followed by a significant input from glacial melt. The catchments in Clusters 1 and 3 also receive significant snowmelt, but the melting starts at least one month earlier in the year than for catchments in Cluster 2. The first peak in SSC for Clusters 1 and 3 is mostly aligned with the first peak of rainfall. The largest difference between the catchments of Cluster 1 and 3 is the amount of glacial melt during late summer. In catchments belonging to Cluster 3, the glacial meltwater peak

perfectly coincides with the second and highest SSC peak of the annual SSC regime. In catchments belonging to Cluster 1, this increase in SSC during peak glacial melt is also visible, but much weaker than for those in Cluster 3. In terms of annual peak discharge (Figure 3g-i), catchments in Cluster 1 show peak discharges approximately simultaneously with the peak SSC, while peak discharge for catchments in Cluster 3 is out of phase with the SSC peak. The spatial distribution of the different catchments across the Alps (Figure 4i) supports the above findings. Catchments belonging to Cluster 2 are mainly located in

the higher alpine region (with one exception located in eastern Austria). The catchments of Cluster 3 follow a clear line from east to west and are bounded to the north and south by the less mountainous catchments of Cluster 1.

The magnitude of the SSC regime, although not used as such to split the catchments into the above-mentioned three clusters, has a moderate correlation (Pearson correlation coefficient >0.5) with a number of catchment characteristics (Figure S7 in the Supporting Information). The median SSC has a moderate positive correlation with the mean daily snowmelt (0.68), mean

daily snow cover (0.67), catchment elevation (0.64), runoff ratio (0.60) and sand fraction (0.58), fraction of precipitation that falls as snow (0.57), fraction of glacier coverage (0.55) and mean daily glacial melt (0.50). A negative correlation is found between the median SSC and the fraction of forest cover (-0.64), mean daily soil moisture (-0.58), fraction of silt and clay (-0.58) and the mean daily air temperature (-0.58). These findings suggest that higher median SSC values should in general be expected in higher elevated catchments that have a large contribution of snow and ice.

## 3.3 Temporal and spatial occurrence of the nine event types

In total, we extracted 2398 extreme SSC events, of which rainfall is by far the most dominant driver. In total, 85 % of the events are purely caused by either high- or low-intensity rainfall (1547 events (64.5 %) and 494 events (20.6 %)) (Figure 5a and c). Those rainfall-driven events occur all year round, but with a higher frequency during summer. Snowmelt-driven events are restricted to the melting season and occur only during late spring and autumn, and their timing is also dependent on catchment

elevation. The same is valid for events that are caused by glacial melt, which only occur during July and August, when the glaciers have become free of snow and become prone to melting. In total, 9.6 % of the events are (partly) snowmelt driven and only 4.6 % are (partly) glacial melt driven.







**Figure 4.** Panels (a-h) illustrate eight static catchment characteristics and their distribution across the three different clusters and within each cluster (the internal distribution is visualised by the boxplots). Panel (i) maps the spatial location of the (nested) catchments over Switzerland and Austria.

The proportion of snow- and glacial melt-dominated events is significantly higher for high-elevation and partially glacierised catchments, which belong to Cluster 2 based on the annual SSC regime clusters. In these catchments, snow- and glacial melt-dominated events account for almost 40 % of all events (Figure 5b). In contrast, lower elevation and larger catchments (Clusters 1 and 3), show many fewer melt-dominated events (< 10 % out of all events).



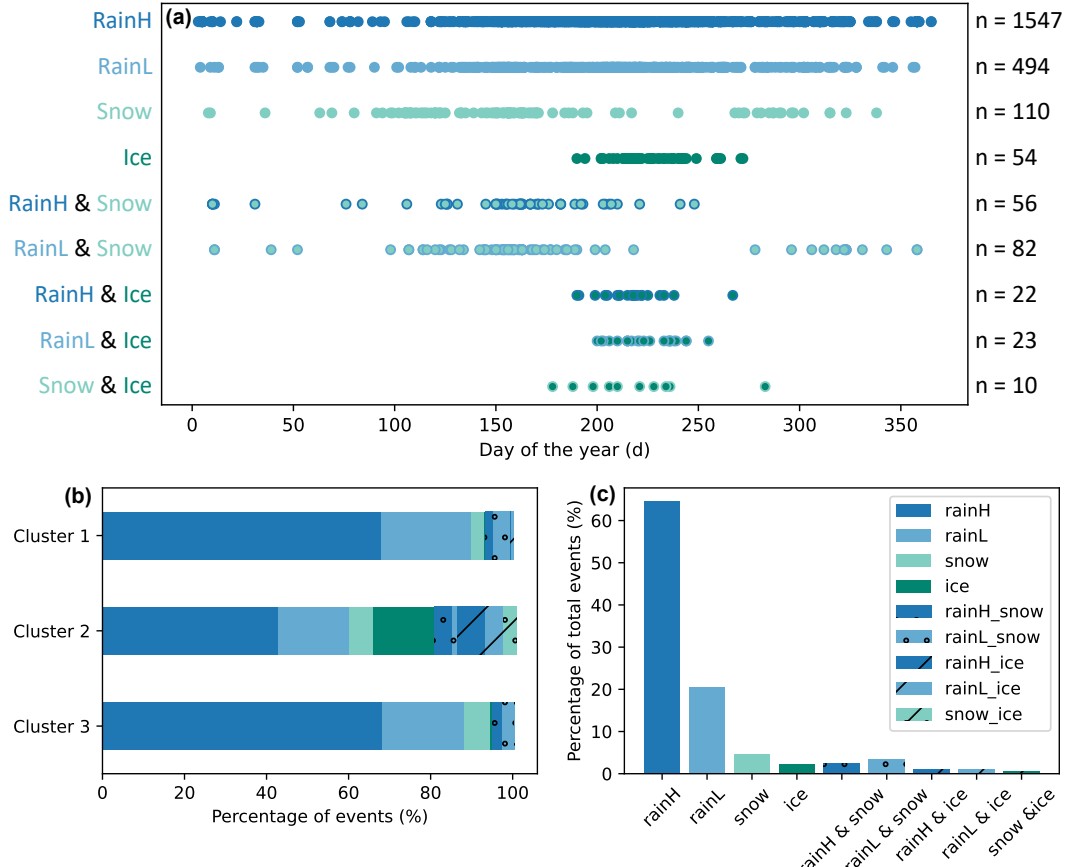

**Figure 5.** (a) Seasonality of occurrence of the nine event types: high intensity rainfall (RainH), low intensity rainfall (RainL), snow melt (Snow), glacial melt (Ice), or a combination of two of the above mentioned transport processes. The total number of events is 2398. (b) Proportion of event types for each of the three annual SSC clusters. (c) Percentage of events per event type compared to the total number of observed events.

### 3.4  Event type characteristics

The event characteristics, such as the event maximum SSC and area-specific suspended sediment yield (sSSY), vary greatly among the different event types (Figure 6 and Figure S8 in Supporting Information). Nine out of the ten most extreme events, with the highest peak SSC, occurred in summer and are caused solely by high-intensity rainfall (peak SSC = 118 g L$^{-1}$ and sSSY = 110 t km$^{-2}$, averaged over the 10 most extreme events) (Figure 6a). Similarly, also the events with the highest sSSY are caused by high-intensity rainfall (Figure S8a in Supporting Information). However, when we look at the average of all events per event type (ignoring the outliers), other event types result in higher peak SSC or sSSY (Figure 6j and Figure S8j in Supporting Information). On average, the highest peak SSC values are caused by events with compounding drivers, namely




those that are caused by a combination of glacial melt and high intensity rainfall (median peak SSC of 5.8 g L$^{-1}$). Events caused by glacial melt alone lead on average to the second highest suspended sediment concentrations (median peak SSC of 4.1 g L$^{-1}$) and the highest specific yields (median sSSY of 28 t km$^{-2}$ per event). Other event characteristics, such as the mean duration of events and the event complexity (number of peaks per event) are less informative as they do not clearly stand out for single event types. All events have a relatively short duration of less than one day, with a mean duration of 17 hours.

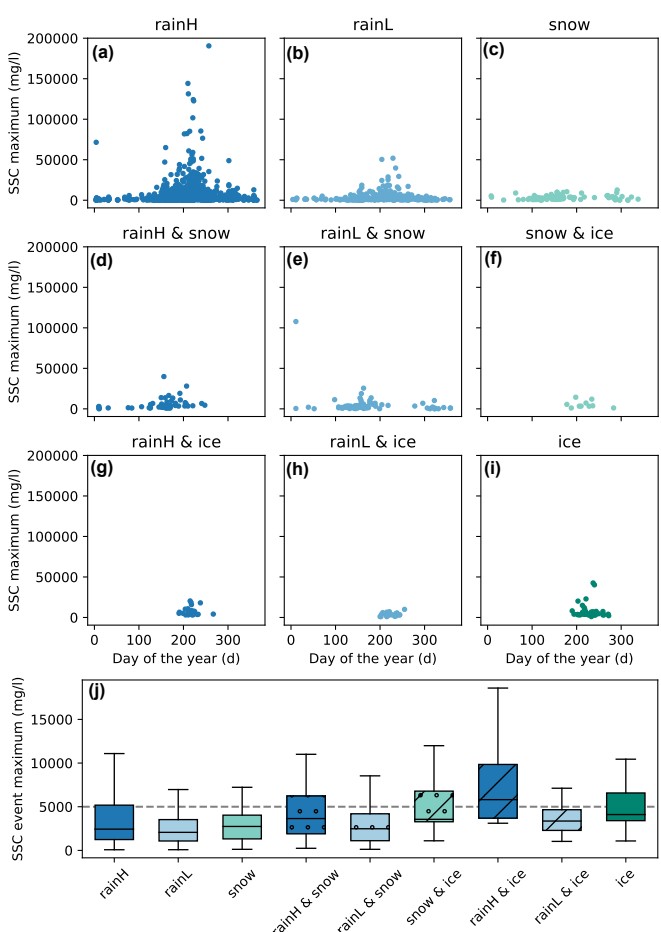

**Figure 6.** Relationship between event magnitude (peak SSC) and generation processes. Panels (a-i) illustrate the variation of peak SSC (including outliers) over time per event type. Panel (j) illustrates the distribution of peak SSC (without outliers) for each event type, with the median value represented by a horizontal grey line.

In addition to the characteristics of the events, we also considered the antecedent conditions, such as the liquid volumetric soil moisture and snow cover, during the two days prior to the event. In general, the majority of events occurs when snow cover is minimal or absent (Figure 7a). However, a more detailed examination of the various event types provides additional insights (Figure S9 in Supporting Information). Events that are partly or entirely driven by glacial melt occur only when the snow cover



has disappeared or is at least smaller than 30 %. In contrast, snowmelt events require the presence of a snow pack, and extreme
SSC events occur under both limited and extensive snow cover. Events driven by a combination of rainfall and snowmelt are
most prevalent when the snow cover is between 10 % and 30 %. However, some particularly severe events have been observed
with a snow cover of 40 % to 70 %. Despite the occurrence of some of the most extreme events with exceptionally high peak
SSC values under conditions with minimal snow cover, no detectable pattern or relationship between snow cover and SSC
could be identified.

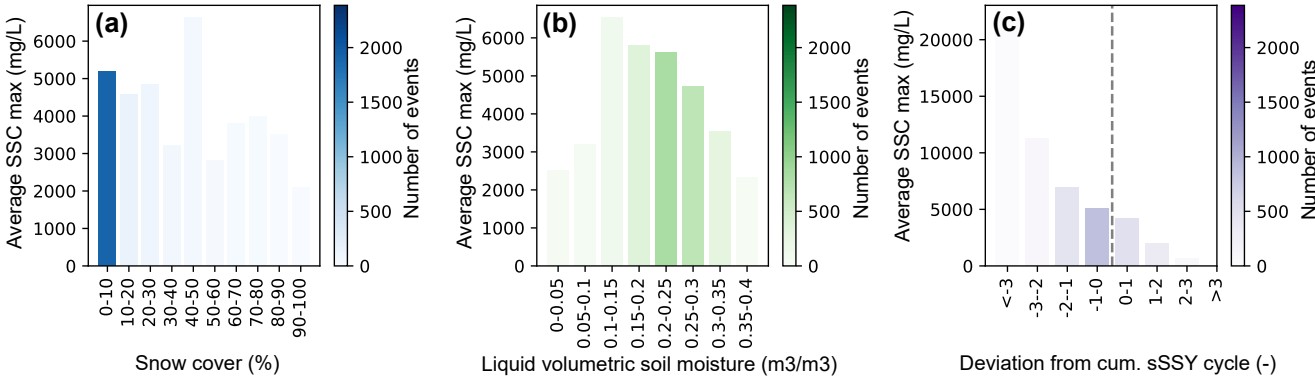

**Figure 7.** Antecedent conditions prior to the extreme SSC events are given for (a) snow cover, (b) liquid volumetric soil moisture, and (c) the
deviation of the daily cumulative sSSY from the annual mean cumulative sSSY regime. Per interval, the bars show the peak SSC averaged
over all the events that belong to each of the intervals. The colour saturation is an indication for the number of events that belong to each
interval. Dark colours mean that most events belong to that interval. Snow cover prior to the event can vary from 0 to 100 %. The liquid
volumetric soil moisture generally varies from 0 to 0.4 $m^3$ $m^{-3}$. The deviation from the cumulative sSSY regime can either be negative
(indicating more sediment availability than usual) or positive (indicating potential sediment depletion).

The majority of events occurs when liquid volumetric soil moisture lies between 0.2 and 0.3 $m^3$ $m^{-3}$ (Figure 7b). The
maximum liquid volumetric soil moisture typically ranges from 0.2 $m^3$ $m^{-3}$ in sandy soils to 0.4 $m^3$ $m^{-3}$ in clay soils. A more
detailed overview per event type is visible in Figure S10 in the Supporting Information, which shows some discrepancies in the
relation between peak SSC and soil moisture between the event types. Glacial melt events predominantly occur under liquid
volumetric soil moisture values of 0.2-0.25 $m^3$ $m^{-3}$, with occurrences at lower or higher soil moisture levels being rare. In
high-elevation and partly glacierized catchments, where glacial melt events are most frequently observed, the soil is typically
composed of a higher proportion of sand, which may contribute to the observed lower liquid volumetric soil moisture. In the
case of rainfall-driven events, the most severe events, with the highest peak SSC, take place under antecedent soil moisture
values between 0.1-0.2 $m^3$ $m^{-3}$, which implies rather dry conditions. The statistical analysis revealed no significant correlations
between soil moisture and SSC.
The proxy for catchment memory and sediment availability, designed as the deviation of the long-term cumulative sSSY
from the mean annual cumulative sSSY regime, shows that more (severe) events occur when the deviation is negative (Figure




7c). A negative deviation indicates a situation in which the transport of sediments during the period prior to the event was below the expected level, suggesting the presence of abundant sediment sources. More negative deviations result in larger observed peak SSC. Similar patterns are observed for rain and snow dominated events, an exception being (partly) glacial melt

events (Figure S11 in Supporting Information). A significant number of glacial melt events occur when the deviation is slightly positive (Figure S11f-i in Supporting Information).

## 4    Discussion

### 4.1    Mean SSC regimes

Our results show that the annual SSC regimes can be divided into three clusters (Figure 3a-c), which are similar in terms of

static attributes (Figure 4a-h) and the seasonal regimes of snow- and glacial melt (Figure 3d-i). This suggests that, in addition to precipitation, the amount of glacial melt and the magnitude and timing of snowmelt strongly control the shape of the annual SSC regime (number of peaks) and the timing of SSC peaks (Figure 3d-i). This is particularly evident when examining catchments in Cluster 3, for which the occurrence of the glacial meltwater peak perfectly coincides with the second and highest SSC peak in the annual SSC regime (Figure 3c and f). The importance of snow- and glacial melt is confirmed by the analysis of the static

catchment characteristics (Figure 4a-h), which shows that characteristics that influence the magnitude and timing of snow- and glacial melt, such as the presence of glaciers (even when covering a small part of the catchment area), catchment elevation, the proportion of precipitation falling as snow, and catchment area, are most important for explaining seasonal sediment dynamics. The importance of meltwater for SSC has already been highlighted by other studies: Costa et al. (2018b) showed that glacial melt produces the highest SSC per unit runoff, Buter et al. (2022) showed that glacial melt results in even higher sediment

concentrations than snowmelt, and Li et al. (2024) showed that the suspended sediment yield of glaciated catchments can be an order of magnitude greater than that of glacier-free catchments. The high concentration of suspended sediment in glacial meltwater can be explained by the importance of glacier forefields as sediment sources and temporary storage. Glacial melt can release sediments that are produced by glacier abrasion on the bedrock and stored in recently deglaciated moraines (Moore et al., 2009; Mancini et al., 2024). These sediments get transported in the subglacial channels, which need time to develop at

the start of the ablation season before they become very effective (Swift et al., 2005). In particular, mainly areas above 2500 m.a.s.l., which are characterized by glacier tongues, bare rock surfaces and recently deglaciated areas, are crucial for sediment generation in high-altitude catchments (Schmidt et al., 2022). A sudden release of suspended sediments has been observed when areas above 2500 m become snow-free and the sediments are no longer protected from mobilization by the snow cover (Schmidt et al., 2022).

In addition, our results show that the relationship between SSC and discharge is not straightforward because the annual discharge regime is completely out of phase with the SSC regime in some catchments (Figure 3), in particular in catchments that belong to Cluster 3 and are characterised by significant glacial melt input during late summer. Discharge has long been considered one of the key predictors of SSC, and the relation between discharge and suspended sediment concentration forms the basis of the well-known sediment rating curves and hysteresis analyses (Walling, 1977). However, other studies have




recently shown that this relationship has been overvalued (Costa et al., 2017; Skålevåg et al., 2024). Sediment rating curves are characterised by a large variability of observations around the regression curve, which can span one or several orders of magnitude, and they tend to underestimate (overestimate) SSC during high (low) discharge (Walling, 1977; Horowitz, 2003; Asselman, 2000). One of the major limitations is that the Q-SSC relationship does not explicitly address the sources of sediment and their activation by different hydro-climatic forcing. Costa et al. (2017) have shown that a sediment source perspective using

hydro-climatic forcing (rainfall, snowmelt and glacial melt) is more appropriate to explore sediment dynamics than traditional discharge-based rating curves. Their findings are consistent with our findings, as our results suggest that the origin of water entering the river system is more informative to understand the mean annual SSC regime than the absolute discharge magnitude on its own.

Even though forcing and transport processes were expected to be among the key determinants of the annual SSC regime,

it is still surprising that static catchment characteristics related to sediment availability, spatial connectivity, geology and soil type did not appear to be more important for explaining the variation in the annual SSC regime among catchments (Figure S6). The catchments belonging to the three clusters of annual SSC regimes did show some mutual differences in the fraction of clay, silt and sand, and we observed a slightly larger fraction of highly erodible geology in catchments of Cluster 1 compared to those in Cluster 3. However, these differences were small and they did not significantly affect the timing and shape of the

SSC regime. We did observe some correlations between a few static characteristics and median SSC, but again characteristics related to snowfall, snow cover, ice and catchment elevation were more important descriptors of SSC behaviour than geology-related characteristics. While SSC is controlled by the depletion and replenishment of different sediment sources over longer time scales (Doomen et al., 2008), we hypothesise that these processes were found to be less relevant in our study because our analyses focused on the mean annual scale.

Looking at the spatial variability of median SSC and sSSY values across catchments, we found that values can vary greatly across locations (Table 2). Median SSC values range from 2.11 mg L$^{-1}$ to 229 mg L$^{-1}$ per catchment, while the mean annual sSSY varies from 19 t km$^{-2}$ yr$^{-1}$ to 1226 t km$^{-2}$ yr$^{-1}$. These values are in line with those previously described in literature, where sSSY values vary between 100-1000 t km$^{-2}$ yr$^{-1}$ as mountainous regions are generally characterised by high sediment yields (Blöthe and Hoffmann, 2022; Vanmaercke et al., 2011; Bogen, 2008; Borrelli et al., 2014; Hinderer et al., 2013; Mano

et al., 2009). In addition, Panagos et al. (2015) predicted a mean soil loss rate of 527 t km$^{-2}$ yr$^{-1}$ for the Alpine climate zone (Alps, Pyrenees, and southern Carpathians), due to the combined effect of rainfall erosivity and topography. One of the highest sediment discharge rates in Europe was recorded in the Italian Central Apennines, with a specific yield of 3235 t km$^{-2}$ yr$^{-1}$ (Borrelli et al., 2014), due to very active geomorphological processes such as gully, rill, bank and channel erosion and re-entrainment of landslide sediments. Such extremely high levels have not been recorded in our catchments.

## 4.2 Extreme SSC events

From the event classification, we conclude that rainfall is by far the most dominant driver of extreme SSC events (Figure 5c), which shows that slightly different processes are important for controlling extreme events compared to annual SSC regimes. In total, 85 % of the events are purely caused by either high- or low-intensity rainfall (1547 events (64.5 %) and 494 events





(20.6 %)) (Figure 5c). This dominant role of precipitation in causing extreme SSC has also been highlighted in other studies
(Schmidt et al., 2022; Mano et al., 2009; Lana-Renault et al., 2007; Blöthe and Hoffmann, 2022). The process of rainfall-driven splash erosion, which is followed by the generation of surface overland flow on hillslopes, results in the delivery of fine sediment to the river network. Given that rainfall affects a much larger area than glacial melt or even snowmelt, and occurs at higher intensities, it was anticipated that rainfall would emerge as the most dominant driver. On the other hand, for snow- and glacier-dominated catchments the number of snow- and glacial melt events was larger and could make up to 40 % of
all events in these catchments. This is consistent with the existing knowledge on runoff generation processes in mountainous snow-dominated catchments, where snow- and glacial melt are significant contributors to overland flow and river discharge (Muelchi et al., 2021).

From the comparison of mean SSC peak values among the different event types, we conclude that events that are partly or entirely dominated by glacial melt result in relatively high SSC (Figure 6j). These findings are in line with existing literature
stating that glacial melt has a significantly higher suspended sediment input per unit of runoff compared to snowmelt and rainfall (Buter et al., 2022; Costa et al., 2017, 2018b). Events with compounding drivers, specifically those resulting from the interaction of glacial melt and high-intensity rainfall, show the highest peak SSC values on average (median peak SSC of 5.8 g L$^{-1}$ and mean peak SSC of 7.9 g L$^{-1}$). However, the 10 highest SSC values in our data set are predominantly associated with high-intensity rainfall events during the summer season, with values reaching up to 118 g L$^{-1}$. Although these observed SSC
values are extremely high, concentrations in the same order of magnitude have been found in other alpine catchments (Mano et al., 2009; Schmidt et al., 2022). Actually, a significant amount of 573 extreme SSC events (24 % of all extreme events) in our data set causes SSC peak values above 5 g L$^{-1}$, which are very likely to be harmful or lethal to a large number of aquatic species, including trout, salmon, and aquatic invertebrates (Newcombe and Macdonald, 1991; Kemp et al., 2011; Collins et al., 2011). Such high levels of SSC also degrade drinking water quality, lower the transparency and limit the penetration of sunlight into
the water, lead to a degradation of the habitat quality for spawning fish, and clog the gills of fish and other aquatic organisms, which can lead to death (Kemp et al., 2011).

On average, the highest event sSSY can be found for events that are driven by glacial melt (46 t km$^{-2}$) or a combination of glacial melt and high-intensity rainfall (37 t km$^{-2}$) (Figure S8 in Supporting Information). During such events, it is likely that large volumes of sediment, produced by glacial erosion and temporarily deposited in proglacial systems, get mobilized and
transported downstream by high-intensity precipitation (Li et al., 2024). Considering a mean soil loss rate of 527 t km$^{-2}$ yr$^{-1}$ for the Alpine region (Panagos et al., 2015), this means that 10 to 13 of those events (generally with a mean duration of less than one day) would be able to account for the entire annual sediment yield of one single catchment. This highlights the importance of those rare but extreme events for annual soil loss by water erosion. Similar results have been found for the Vent catchment in Ötztal, where individual summer rainstorm events can account for up to 26 % of the annual sediment yield within just over
24 hours (Schmidt et al., 2022). In contrast, Blöthe and Hoffmann (2022) reported that in the upland tributaries of the Rhine in central Germany, without the presence of glaciers, the 10 largest events were only able to transport 8 to 25 % of the total annual suspended sediment load.





To be more precise, the event sSSY would ideally be compared to the mean annual sediment yield of the specific catchment under consideration, because the annual sediment yield can vary considerably between catchments (Table 2). After correcting

for the catchment-specific annual yield, we found that a number of events that contributed disproportionately to the annual sediment yield were related to two significant storm and flood events that occurred in Europe: the large flooding in Central Europe in early June 2013, with precipitation exceeding 300 mm over four days on the northern side of the Alps resulting in large flooding in the Upper Danube basin (Blöschl et al., 2013), and the storm "Vaia", which hit southern Austria on October 28th in 2018 and was characterized by extreme accumulated precipitation of up to 850 mm in three days (Giovannini et al.,

2021). Our results show that some headwater catchments transported more than four times the average annual sediment yield in only 6 to 8 days. Further downstream, the specific suspended sediment yields of these flood events were still able to account for up to 150 % to 280 % of the annual sediment yield. Such high sediment yields are only possible in combination with landslides and river flooding, when all the sediment within and near the river bed gets activated, which both have been recorded for the 2013 and 2018 event (Blöschl et al., 2013; Giovannini et al., 2021). While events with a high sediment yield play a significant

role for soil loss within a catchment, a high sediment yield (and high discharge) does not necessarily lead to a high sediment concentration (Zheng et al., 2021).

The relationship between antecedent conditions and extreme suspended sediment concentration (SSC) events is complex. Notably, some of the highest peak SSC and specific suspended sediment yields (sSSY) occurred during periods of high soil moisture and limited snow cover (Figure 7), as saturated soils are more prone to landslides and snow free soils are less protected

from erosion (Godt et al., 2009; Hamshaw et al., 2018). However, we also noticed that glacial melt occurs only when the snow cover has disappeared or is at least smaller than 30 %. This could be a model artefact, as it is consistent with how glacial melt is represented in the PCR-GLOBWB 2.0 model, which assumes that glaciers only become vulnerable to melting when they are no longer covered (and protected) by snow. In terms of soil moisture, moderate rainfall during previously wet conditions can also lead to high runoff without a corresponding increase in SSC due to dilution effects (Lana-Renault et al., 2007). When

the entire catchment contributes to discharge, sediment concentration may not rise if sediment input is proportional to water input. Conversely, dry soils can also be prone to erosion at the onset of rainfall events due to hydrophobicity, splashing, and surface sealing (Zambon et al., 2021). Our observations indicate that high-intensity rainfall events often result in more severe SSC under relatively dry conditions compared to wet ones (Figure S10 in Supporting Information). These contrasting signals complicated efforts to establish a clear correlation between SSC and soil moisture (Figure 7b).

Our proxy for catchment memory, designed based on the long-term deviation of cumulative sSSY, shows a positive relationship between the availability of sediment sources and event peak SSC. We observe more events with higher peak SSC when sediment resources are still abundant (Figure 7c). This is in line with our expectations, as the presence of abundant sediment sources creates favourable conditions for the occurrence of extreme SSC events. It should be noted, however, that no relationship was visible when the catchment memory was reset each winter, implying that sediment stores build up and empty over

several years. In our case, deviations in sediment storage developed and were tracked throughout the time series (10-12 years).

The combination of greater sediment availability and sufficient forcing (e.g., high-intensity rainfall or snowmelt) results in larger observed peak SSC. The only exception are those events that are (partly) driven by glacial melt. These events also




occur frequently during conditions in which more sediment than usual has already been transported prior to the event (Figure S11f-i in Supporting Information). One potential explanation for this is that events driven by glacial melt rely on a different
sediment source compared to those driven by snowmelt and precipitation, namely sediments from the glacier and its forefields (Mancini et al., 2024). Consequently, the transportation of sediments during glacial melt is independent on the availability of sediments in other areas within the catchment. However, this is inconsistent with the fact that this method makes use of the catchment specific annual cumulative sSSY regime, which already accounts for the different sediment sources within a catchment. Possibly, these events occurred during an exceptional period of glacier retreat or glacier motion that exceeded the
annual average, as rapid glacier retreat and areas of recently deglaciated terrain reveal effective sediment sources (Moore et al., 2009; Swift et al., 2005).

## 4.3 Limitations and generalizability

The results presented are associated with uncertainties, which result from a combination of uncertainties inherited from the underlying data and methodology. One important data-related limitation is the uncertainty in the turbidity-SSC relationship.
Despite the many efforts by the Swiss and Austrian hydrographic services in collecting bi-weekly manual SSC samples, it is difficult to reconstruct the relationship between turbidity and SSC reliably. The non-linear regression model attempts to represent the relation of SSC with turbidity, but the actual variability in the relationship between turbidity and SSC will likely be more stochastic as it is influenced by the varying characteristics of the sediments in suspension (particle size, shape, and density) (Gippel, 1995) and is also strongly dominated by mass wasting (Battista et al., 2022). Nevertheless, the turbidity-SSC
relationship is currently the most applied and accepted method to derive high-resolution SSC time series described in the literature, and has been used by Costa et al. (2018a); Pellegrini et al. (2023); Stott and Mount (2007); Thollet et al. (2021), among others. Because of the uncertainty in the turbidity-SSC relationship, we removed a few implausible outliers and decided to aggregate the data from 10 and 15 minutes to mean hourly values. Before analysing the annual SSC regimes, we had to smooth the time series over a 30-day window to account for the relatively short time series. A further limitation of this study
is the limited availability of SSC data, which has resulted in a relatively small sample size. At the same time, comprehensive large-sample studies are scarce. By analysing 38 rivers across the Alps, we believe that our analyses capture the most important processes and their variability in space and time.

Other uncertainties can be related to the methodology, e.g. the hierarchical clustering approach, the definition of extreme events, and the event classification scheme developed. For the clustering of the annual SSC regimes, we applied a hierarchical
clustering algorithm. A known disadvantage of this clustering approach is that the selected indicators can greatly influence the final clusters. However, a comparison with clusters derived by $k$-means clustering showed that the clusters identified are relatively stable independent of the choice of the clustering technique. We chose to select indicators that were able to capture the shape of the SSC regime, rather than the variation in magnitude or the synchronicity with the discharge regime. This choice is reflected in the clusters identified, which differ mostly in terms of the number and timing of the main peaks.
For the detection of extreme SSC events, we applied a peak-over-threshold approach which is widely used (Skålevåg et al., 2024; Hamshaw et al., 2018; Haddadchi and Hicks, 2021; Blöthe and Hoffmann, 2022). At the same time, the definition of




extremes necessarily entails some degree of subjectivity and arbitrary decision-making. Instead of using the 99th percentile threshold, a lower threshold could be selected (for instance at the 90th percentile). This would result in the extraction of more but less extreme events. Furthermore, the methodology employed to define the start and end of an event is inherently subjective.

The definition of the start of extreme SSC events is rather straightforward as it is characterised by a sudden and pronounced increase in SSC, that is easily detectable by a change in the slope. However, determining the end of the event was more challenging. We tested and compared three different methods within the framework of a sensitivity analysis, including the use of the slope, another fixed threshold, or the decrease of SSC relative to the SSC peak (Figure S5 in Supporting Information). The duration and the calculated sSSY of the event varied slightly depending on the selected method. However, the sensitivity

analysis demonstrated that our primary results and conclusions remain unaffected by such changes in the definition (Table S3 in Supporting Information).

The classification scheme that we presented here is relatively straightforward to apply and therefore explainable. It requires only few inputs (hourly precipitation and daily snow- and glacial melt) and is based on the relatively well-understood processes of sediment transport, while more complex processes of sediment availability are ignored for simplicity. Our event classification

relies on our input data and as the snow and glacial melt data are evaluated over the larger Alpine domain, they might be more uncertain in smaller catchments and on shorter time scales. However, general melt patterns are well-represented. Since our classification focuses on the dominant processes, it is less sensitive to smaller-scale uncertainty and increases our trust in our classification. We show that the method is applicable to catchments of different sizes and the absence of glaciers does not have a detrimental impact on the results. Given that the threshold for differentiating between dominant transport processes

is set at 1/3, one would expect a considerable number of precipitation-snowmelt events in catchments without glacial melt, but with substantial snowmelt contributions. However, our analysis indicates that this is not a concern, as the number of precipitation-snowmelt events is relatively limited even in non-glaciated catchments. Our event classification differs from existing classification techniques, such as those proposed by Millares and Moñino (2020) and Skålevåg et al. (2024), which both rely extensively on the SSC-discharge relationship. In fact, the analysis of the annual SSC regimes has led us to conclude

that discharge is a poor predictor of SSC in some catchments, thereby supporting the decision to select only hydro-climatic drivers for the classification scheme.

Our choice of methods and the use of a large catchment sample all contribute to the generalisability of our results. This study not only improves our understanding of the complex hydrological-sedimentary response in the studied catchments, but due to its generalisability, this knowledge may also be transferred to unmonitored river systems in the same region. The selected

methods, such as the clustering of SSC regimes and the event classification scheme, are readily transferable to other catchments and regions, thus opening the door to further large sample studies focusing on SSC.

### 4.4 Implications of findings and outlook

The importance of snow- and glacial melt for both seasonal and event SSC dynamics, suggests that climate change might affect sediment concentrations and fluxes in Alpine rivers in the future. There is evidence that glacier retreat results in a

potential increase in suspended sediment concentrations and fluxes, and this effect can last for decades and centuries in small



alpine watersheds (Moore et al., 2009; Buter et al., 2022). However, a long-term decline in suspended sediments delivered from glacial sources is anticipated under future climate change scenarios since the area of glacier cover and glacier erosion will decrease and glacier forefields will stabilize (Moore et al., 2009; Mancini et al., 2024). Future projections of sediment export for two catchments in Ötztal suggest a decrease in sediment export already in the coming decades, and imply that 'peak

sediment' has already passed (Schmidt et al., 2024). This is in line with findings by (Freudiger et al., 2020) who concluded that glacier peak water was already reached in most of the catchments in the Swiss Alps in the past decades and will be reached in all catchments during the first half of this century, independent of the emissions scenario. In addition, also for non-glaciated catchments, such as the Illgraben valley, one of the most geologically unstable regions of Switzerland, studies suggest a future decline in sediment transport (Hirschberg et al., 2021). Although future climate conditions are expected to favour an increase

in the sediment transport capacity in this valley, a reduction in sediment supply produced by frost weathering may limit the debris-flow activity and the actual sediment transport.

In addition to alterations in glacial coverage, future changes in snow seasonality, reduced snow accumulation and coverage are also expected to influence the sediment availability in catchments and the concentrations in rivers (Maruffi et al., 2022). When sediment sources are covered by snow they are protected from erosion and not connected to the river system, resulting in

a reduction in SSC during winter (Schmidt et al., 2022). Consequently, shorter periods with reduced snow cover will lead to an increased connectivity of potential sediment sources to the river, which may lead to higher SSC. Nevertheless, we hypothesize that an extended period during which sediments can be transported may also cause a depletion of sediment sources, resulting in lower SSC later in the year.

Likewise, the impact of projected future precipitation on changes in sediment concentration remains incompletely under-

stood, particularly in combination with sediment availability. Heavy and extreme precipitation events in the summer months will occur with greater frequency and intensity (Wood and Ludwig, 2020; Martel et al., 2020), and such occurrences have the potential to cause significant erosion and an elevated annual yield (Schmidt et al., 2024). However, as a consequence of dilution, high precipitation and high discharge do not necessarily result in the highest suspended sediment concentrations (Lana-Renault et al., 2007). While these alterations in the contribution of rain, snowmelt, and glacial melt will result in a shift in the timing

of runoff and discharge seasonality, their impact on the SSC regime remains uncertain. Our observation-based insights into the relationship between melt processes and SSC suggest future changes in SSC behaviour. However, targeted modelling and field experiments are needed to better understand the whole process chain from weathering to erosion, sediment storage and sediment transport, and to make reliable projections of future SSC.

## 5   Conclusions

In this paper, we identified the main factors influencing the spatial and temporal variability of the annual SSC regime and the occurrence of extreme SSC events in 38 catchments in the Alpine region. Our results demonstrate that the annual SSC regime and extreme SSC events in small mountain catchments are substantially influenced by snow and ice, in contrast to low-elevation and large catchments. The presence of glaciers and the magnitude and timing of snowmelt are important factors influencing



the annual SSC regime and controlling the timing of peak SSC, while geology- and soil-related catchment characteristics
and the annual discharge regime appear to have a smaller influence. Furthermore, we present a novel classification scheme
to categorise extreme SSC events into nine different event types. Our analysis of 2,398 extreme SSC events indicates that
rainfall is the primary driver of SSC extremes, accounting for 80 % of all events. This shows that slightly different processes
are important for controlling extreme events than for the annual SSC regime. However, in high-elevation and partly glaciated
catchments, up to 40 % of the events are still attributable to snow- and glacial melt. Events with compounding drivers, namely
glacial melt and high-intensity rainfall, result in the highest sediment concentrations and area-specific yields. Events driven by
glacial melt have a specific yield of, on average, 46 t km$^{-2}$, which means that ten of these short-term extreme events account for
the transport of the total annual sediment yield of a catchment. Moreover, a considerable proportion of the extreme SSC events
(24 % of the total) resulted in peak SSC values exceeding 5 g L$^{-1}$, which can have detrimental effects on aquatic ecosystems.
These findings highlight the importance and impact of such events on the water quality in Alpine rivers and give an indication
of soil loss due to water erosion.

*Data availability.* The shapefiles of all catchments, including static catchment characteristics, annual regime data and event data are available
through HydroShare according to the FAIR data sharing principles: van Hamel, A. (2024). Suspended sediment concentration in Alpine rivers
- annual regimes and extreme events, HydroShare, http://www.hydroshare.org/resource/8ec269a1e512434c9acb76b74025e8f7

*Author contributions.* AVH developed the general idea and conceptualized the study with MiB and PM. JJ provided the snowmelt and glacial
melt data that was used as input. AVH compiled the data and performed the analysis. The first draft of the paper, including all the figures,
was written by AVH with contributions from all co-authors. MiB, PM, and JJ revised and edited the document.

*Competing interests.* The authors declare that they have no conflict of interest and that MIB is an editor with HESS

*Acknowledgements.* The authors would like to thank the Swiss Federal Office for the Environment and the Austrian Hydrographic Service
for data provision.



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
