# Peer review of "Suspended sediment concentrations in Alpine rivers: from annual regimes to sub-daily extreme events"

_EGUsphere, 2024_

## Author Response (AR1)

*Dear Prof. Saco,*

*Thank you very much for giving us the opportunity to revise our manuscript. We highly appreciate the constructive comments and specific suggestions by the two reviewers on how to improve our manuscript. The revised version of our manuscript comes with the following minor changes: (1) all specific comments from the reviewers have been addressed, as previously outlined in our response to the reviewers. A comprehensive, point-by-point overview is provided below. (2) Our colleague Joren Janzing has recently made some adjustments to the glacier module of the PCR-GLOBWB 2.0 model. This resulted in minor changes to the simulated gridded daily ice melt data, which was used as the input for our study. To maintain integrity, some figures (Figs. 3-6) and the online data provided via Hydroshare have been updated accordingly. It is important to emphasize that these changes had no impact on the conclusions of the study.*

*We hope that you find the revised version of our manuscript suitable for publication and thank you very much for your re-consideration.*

*Best regards,*

*Amber van Hamel,*

*On behalf of Manuela Brunner, Joren Janzing, and Peter Molnar*

**Point-by-point reply to the comments**

*Please note: Newly added text is written in italics. For each adjustment, we include the line of text in the manuscript where this change is made (see the manuscript version with changes marked in red).*

==RC1: 'Comment on egusphere-2024-3985' **Panayiotis Dimitriadis, 08 Feb 2025**==

In this study, the Authors investigate the principal factors that influence the spatiotemporal variability and frequency of extremes of Suspended Sediment Load (SSL) in the Alps; please see some minor and one major issue that I hope they can be of help to the Authors:

1) Please mention in the Abstract the hierarchy of the identified impacts of "the catchment elevation, the start of the melt season, and the presence of glaciers.".

**Reply**:  Unfortunately, our method does not allow us to rank impacts, but it ranks MST-indicators. Hierarchical clustering is applied to show us which of the MST-indicators were the most influential in clustering the catchments and their SSC regimes. These appeared to be the number of peaks and the timing of the main SSC peak. The next step was to interpret the three clusters and explain the spatial variation between the SSC regime types. This step involved comparing z-scores for different characteristics across the various clusters but is somewhat subjective as it does not generate a defined hierarchy where one impact or characteristic is clearly more important than another. Therefore, we have left the text unchanged without suggesting the existence of any hierarchy between the impacts.

2) I think the main conclusion that "The analysis of 2,398 extreme SSC events across all catchments indicates that rainfall is the primary driver of SSC extremes, responsible for 80% of all events." may be expected. Please link this to the literature regarding existing SSC hydrological

models that also use rainfall as the primary impact force. Also, please mention in the Abstract and Conclusions about the rest of 20% that rainfall was not identified as the primary factor of SSC events and please give further details on the primary factors and how can these be physically justified.

**Reply**: Thank you for highlighting the need to (a) clarify the link between our finding that precipitation is the primary driver of SSC extremes and the literature regarding existing SSC models and (b) elaborate more on the non-rainfall driven events and how these events can be physically justified in the Abstract and Conclusion.

In the discussion section 4.2, Line 469ff, we elaborate shortly on precipitation being the primary driver of SSC by stating: "This dominant role of precipitation in causing extreme SSC has also been highlighted in other studies (Schmidt et al., 2022; Mano et al., 2009; Lana-Reneault et al., 2007; Blöthe and Hoffmann, 2022). The process of rainfall-driven splash erosion, which is followed by the generation of surface overland flow on hillslopes, results in the delivery of fine sediment to the river network. Given that rainfall affects a much larger area than glacial melt or even snowmelt, and occurs at higher intensities, it was anticipated that rainfall would emerge as the most dominant driver."

We have extended this part by adding the following sentence in which we make the link to SSC modelling: *"Therefore, precipitation is also increasingly considered as a valuable input to various SSC modelling approaches, such as the process-based rating curve of Costa et al. (2018), the extended sediment rating curve of Wolf et al. (2023), and the machine learning model of Aires et al. (2023)."* (Line 468 in the manuscript with marked changes)

We agree that the conclusions and abstract could profit from an expansion of the discussion on non-rainfall driven events, especially because this is one of our main findings. We have added the following text to the Abstract and make similar adjustments in the Conclusion:

"[…] rainfall is the main driver of SSC extremes, responsible for *85% of the events. The remaining events are entirely or partly associated with snow and glacial melt,* which can account for up to 35% of events in high-elevation and partially glaciated catchments." (L9)

And

"[…] *This underscores* the disproportionate influence of meltwater on sediment concentrations in high altitude Alpine rivers*, which can be explained by the significant contribution of meltwater to overland flow and river discharge in combination with the high sediment availability in glacier forefields.* […]" (L10)

3) The Authors use the SSC units "g/L". I think for all the SSL events, and their impacts to the environment and humans, to can be directly compared, the SSL material should be similar; is the SSL material similar in all the examined catchments/rivers in the Alps, and if not, maybe it could be useful to mention the differences in the SSLs.

**Reply**: Thank you for raising this point. In our study, we explicitly focus on SSC events, because it is the concentration rather than the sediment load, source, or composition that matters for the impacts on aquatic ecosystems and water quality (Coffey et al., 2019; Newcombe and Macdonald, 1991). In general, the SSC in "g/L" can be compared well among rivers with different sediment sources and annual sediment loads. However, as the rivers in the Alps do indeed show a large variation in annual median SSC (see Table 2 in our manuscript), and as ecosystems adapt to the local conditions, we decided to use a locally defined 99th percentile

threshold to detect locally relevant extreme SSC events (instead of using a fixed threshold). This means that, at each location and time of the year, the extreme events defined by our methodology are unusual and have a low probability of occurring. Nevertheless, we can still investigate when and where (independent of the river average) events with an extremely high value occur, for example events with a concentration of more than 5 g/L.

In addition to the SSC, we also look at the area-specific suspended sediment yield (sSSY) which has been transported during the SSC events. The sediment yield (or load) is also important to describe the impact on the environment and humans, but from the perspective of (actual) soil loss through water erosion. As we state in the discussion in line 503 "[…] the event sSSY would ideally be compared to the mean annual sediment yield of the specific catchment under consideration, because the annual sediment yield can vary considerably between catchments". If we understand your comment correctly, you propose to correct for this by implementing it in our method (instead of only discussing it in the discussion part of our manuscript). We agree that this is a valuable adjustment to make. Therefore, we have replaced Figure S8, where we previously showed the event sSSY over time and per event type, with a new figure where we show the sSSY event fraction in relation to the catchment annual sSSY (see below):

[Figure]

Figure 1 Replacement of Figure S8 in Supporting Information. Instead of the 'Event sSSY' (on the y-axis), we plot the 'Fraction of the event sSSY of annual sSSY (-)'.

Our main conclusions are hardly affected by this change. However, we had to make some adjustments to our text in the methods, results and discussion sections, as shown below:

Method section 2.4.3, at line 275 (new/adjusted text in *italic*): "Event characteristics include event duration, mean and maximum event SSC, specific suspended sediment yield (sSSY) [t km$^{-2}$], and event complexity, defined as the number of peaks in the SSC between the start and end of the event (Figure 1b). *We also calculated the sSSY event fraction of the mean annual sSSY per catchment as an indication of the extreme nature of the event in terms of the sediment load transported.* In addition, we monitored important seasonal-varying antecedent conditions [...]"

Result section 3.4, from line 359 and further (new/adjusted text in *italic*): "The event characteristics, such as the event maximum SSC, the area-specific suspended sediment yield (sSSY), *and the sSSY event fraction of the annual mean sSSY,* vary greatly among the different event types (Figure 6 and Figure S8 in Supporting Information). Nine out of the ten most extreme events, with the highest peak SSC, occurred in summer and are caused solely by high-intensity rainfall (peak SSC = 118 g L$^{-1}$ and *sSSY event fraction = 0.33,* averaged over the 10 most extreme

events) (Figure 6a). Similarly, also the events with the highest sSSY *and sSSY event fraction* are caused by high-intensity rainfall (Figure S8a in Supporting Information). However, when we look at the *median* of all events per event type, other event types result in higher peak SSC (Figure 6j*) or high sSSY event fractions* (Figure S8j in Supporting Information). [...] Events caused by glacial melt alone lead on average to the second highest suspended sediment concentrations (median peak SSC of 4.2 g $L^{-1}$), the highest specific yields (median sSSY of 32 t $km^{-2}$ per event)*, and the second highest fraction of event sSSY (median sSSY event fraction of 0.03).*"

Discussion section 4.2, rewriting of paragraphs 3 and 4 (between line 490 and 508):

"On average, the highest event sSSY is found for events driven by glacial melt (52 t $km^{-2}$) or a combination of glacial melt and high-intensity rainfall (42 t $km^{-2}$ ) (Figure S8 in Supporting Information). During such events, large volumes of sediment produced by glacial erosion and temporarily deposited in proglacial systems are likely to be mobilised and transported downstream by high-intensity precipitation (Li et al., 2024). *If we compare the event sSSY with the mean annual sSSY of the catchment, we see that these extreme events transport on average 2 % of the catchment's annual sSSY. However, some high-intensity rainfall events can transport up to 20 % or 60 % of the annual sSSY.* Similar results were found for the Vent catchment in Ötztal, where individual summer rainstorm events can account for up to 26 % of the annual sSSY in just over 24 hours (Schmidt et al., 2022)

*During some of the most sediment-productive events,* some headwater catchments transported more than four times the average annual sSSY in just 6 to 8 days. *Further investigation showed that these events* were related to two major storm and flood events that occurred in Europe [...]. While these events with very high sediment yield play a significant role in soil loss within a catchment, *a high sediment yield does not necessarily lead to a high sediment concentration as long as it is combined with high discharge* (Zheng et al., 2021)."

4) Regarding soil-moisture and streamflows, did the Authors considered land-use changes during these 10 years of data that could have an impact to both these processes and so, to the measured SSL?

**Reply**: We have not actively included land-use changes in our analysis as land use changes in the Alps are not substantial over this short time scale. However, it is possible that smaller land-use changes have influenced the soil-moisture and streamflow data to a limited degree. We have clarified this in Section 2.2.2. (Methods) by mentioning that land-use changes have not been explicitly considered, and we also touch upon this briefly in the discussion (Section 4.2).

Added to section 2.2.2: *"Because of data scarcity and the relative short time series, we have decided to not explicitly consider the effect of land-use changes, although potential effects of such changes might be indirectly represented in the time series of soil moisture and discharge."* (L181)

Added to section 4.2: *"A final complicating factor, which we did not explicitly consider in this study, is the potential effect of changes in land use on soil moisture, discharge, and erosion. All these contrasting signals have complicated efforts to establish a clear relationship between SSC and soil moisture."* (L518)

5) The Authors use 10 years of observed sub-daily SSL data from 38 gauging stations in Switzerland and Austria, and they consider "time-varying hydro-climatic processes and forcing"; I am concerned that 10 years of data may not be sufficient for the investigation of

spatio-temporal variability, and at least 30 years of data is required due mainly to the long-term persistence climatic driver (also in line to the IAHS scientific view discussed in Montanari et al., 2013) that has been reported to govern key hydrological-cycle processes for the SSL (such as precipitation and streamflow; for example, see a global-scale analysis in Dimitriadis et al., 2021, where medium and strong Hurst exponents have been found for both these processes) and cause intense hydro-climatic variabilities (for example, see the European-scale study by Blöschl et al., 2019).

Blöschl et al., Changing climate both increases and decreases European river floods, Nature 573(7772), pp. 108-111, DOI: 10.1038/s41586-019-1495-6 , 2019.

Dimitriadis et al., A Global-Scale Investigation of Stochastic Similarities in Marginal Distribution and Dependence Structure of Key Hydrological-Cycle Processes, Hydrology, 8, 59. https://doi.org/10.3390/hydrology8020059, 2021.

Montanari et al., Panta Rhei – Everything Flows, Change in Hydrology and Society – The IAHS Scientific Decade 2013-2022, Hydrological Sciences Journal, 58 (6), 1256–1275, doi:10.1080/02626667.2013.809088, 2013.

**Reply**: We agree that trend analyses would require time series longer than 10 years. Here, we do not aim to study such long-term variation but rather variations over shorter time scales that are well represented in the time series available. While longer time series would also have been useful for such an analysis, we prioritized spatial coverage over time series length because we aimed to understand spatial variations in SSC extreme events. We realize that our use of the term 'time-varying' processes might have been confusing. By 'time-varying' we mean those characteristics that (unlike static catchment characteristics) are dependent on time and show strong seasonal variation, e.g., the melt processes, active drainage area, soil moisture, and sediment availability (as described in Section 2.2.2).

We have made some textual adjustments to clarify that by 'time-varying' we mean seasonal changes rather than long-term trends:

- We replaced 'time-varying' with 'seasonal-varying', everywhere in the text.
- We replaced 'temporal' by 'seasonal' where possible, for example:
    - "[…] (i) to quantify spatial and  *seasonal* differences in the annual SSC regime [...]" (L79)
    - "[…] (iii) to explain the spatial and  *seasonal* differences between extreme SSC event types [...]" (L81)
    - "[…] Spatial and  *seasonal* differences in the annual sediment regime" (L185).

*Coffey et al., A review of water quality responses to air temperature and precipitation changes 2: Nutrients, Algal Blooms, Sediment, Pathogens, JAWRA Journal of the American Water Resources Association, 55 (4), 844-868, doi: 10.1111/1752-1688.12711, 2019.*

*Newcombe and Macdonald, Effects of suspended sediments on aquatic ecosystems, North American Journal of Fisheries Management, 11 (1), 72-82, doi: 10.1577/1548-8675(1991)011<0072:EOSSOA>2.3.CO;2, 1991.*

**RC2:** 'Comment on egusphere-2024-3985', Anonymous Referee #2, 14 Feb 2025 reply

Review of „Suspended sediment concentrations in Alpine rivers: from annual regimes to sub-daily extreme events"

The authors investigate median annual suspended sediment concentration regimes for 38 gauges in the European Alps and classify extreme SSC events.

This is an interesting and timely study, and I provide some major and minor comments that should be addressed before this study can be published.

Major comments:

Abstract: Ideally, the abstract would be very easily skimmable – also and especially for an audience that is not experts in the field – to determine quickly whether or not the paper is relevant to them. At its current state, some party are rather difficult to read quickly because of the complicated wording and or long sentences. Consider replacing some words and shortening some sentences. E.g. the sentence starting in line 6: utilizing à using, maybe make two sentences out of this; there is a lot to digest here (the reader has to consider what the "annual median SSC regime" is, what hierarchical clustering is and does and what you use to run it). Also using paragraphs would be helpful for readers to find the structure better.

**Reply**: Thank you for highlighting the need to simplify the abstract. We have revised the abstract by shortening sentences, rephrasing some passages, and making it less technical in the following way (total length = 245 words):

"The occurrence of extreme suspended sediment concentrations (SSC) in rivers can have *negative impacts* on human infrastructure, water quality, and the health of aquatic ecosystems. However, *most* existing studies have focused on the SSC dynamics of individual catchments or single events. Consequently, large-scale patterns of suspended sediment dynamics remain poorly understood. The objective of this study is to identify spatial differences in (1) the seasonality of SSC and (2) the occurrence of SSC extremes in the Alps. For our analyses, we use 10 years of observed sub-daily SSC data from 38 gauging stations in Switzerland and Austria.

We show *that the presence of glaciers, catchment elevation and the onset of the melt season are important drivers of SSC seasonality*. However, slightly different processes are important at the event scale, where rainfall is the *main* driver of SSC extremes, responsible for 85 % of all events. *The remaining events are entirely or partly associated with snow and glacial melt, which can account for up to 35% of the events in* high-elevation and partially glaciated catchments. *This underscores* the disproportionate influence of meltwater on sediment concentrations in high altitude Alpine rivers*, which can be explained by the significant contribution of meltwater to overland flow and river discharge in combination with the high sediment availability in glacier forefields*. A *significant* proportion of the extreme events (24%) resulted in peak SSC values *greater than* 5 g L$^{-1}$, highlighting their potential *to cause significant harm to* aquatic species and *river* ecosystems."

L 97 ff: The novelty kind of doubles with the aims of the study, and this is repeated again in the first methods paragraph (L 104ff), sometimes even with the exact same wording. I think, this can be shortened.

**Reply**: Thank you for pointing out these redundancies. We removed these redundancies by removing the first methods paragraph and adjusting the introduction as follows:

"The novelty of this study is fourfold: (1) We examine key indicators of the annual SSC regime and extreme SSC events  over a large region; (2) We include [...]" (L87)

L 177: "Static characteristics [...] such as [...] mean daily temperature" – I would argue that temperature is not static, and I assume you would agree that there is a trend in temperatures. Thus it is important to state more exactly what you mean here (mean daily temp over which period of time? The 12 or so years you examined? Or the standard 30 years?).

**Reply**: We agree that the use of the word 'static' is somehow confusing and that temperature is not a static variable. Therefore, we have removed the word 'static' in the text and talk about catchment characteristics instead. In the sentence you mention, we have specified the period over which the mean daily temperature was calculated: "*Catchment characteristics* include *physiographic characteristics*, such as elevation, but also hydro-*meteorological characteristics*, such as mean daily temperature *averaged over the available time series length within the period 2009-2023.*" (L159)

L 179: I would argue that maybe the most important characteristic related to sediment availability is the presence or absence of glaciers in a catchment, but it is missing here. Maybe I don't understand where you are going with this?

**Reply**: We agree that the fraction of glacier coverage in a catchment is indeed one of the key characteristics explaining sediment availability. This is why we have included it in the set of catchment descriptors used for our analyses. Thank you for highlighting that this has not been sufficiently clear. We have clarified this by adding *'the fraction of glacier cover'* to the examples of static characteristics that we mention in the text:

"Characteristics related to sediment availability are for example *the fraction of glacier cover*, the slope, runoff-ratio, mean daily discharge, and mean daily precipitation. Characteristics related to sediment availability are for example the land cover types and geology classes that are present in the catchment. Figure S2 and S3 in the Supporting Information provide an overview of all static characteristics that we considered in this study." (L161)

L193: there is no reference given for the actively contributing drainage area. I think it has been used/developed with respect to sediment by Li et al.? And also been used by Schmidt et al.?

**Reply**: Yes, we indeed use a similar definition of active contribution drainage area as introduced by Li et al. (2021) and Schmidt et al. (2022). We clarified the definition of active drainage area in the text and added the missing references:

"An important catchment characteristic that changes per SSC event and affects the sediment availability is the actively contributing drainage area. *We adopt the definitions by Li et al. (2021) and Schmidt et al. (2022) by considering snow-free areas to be potentially erodible.* Other time-varying catchment characteristics [...]" (L177)

L230: You tested using a different method and got similar clusters, which is great. Have you also tested for the importance of individual gauges, i.e. removing some gauges and running the clustering to see whether you still get the same results? It would be good to know how sensitive the result is to the choice of stations.

**Reply**: Thank you for this great suggestion based on which we have tested the sensitivity of our clusters to the removal of individual gauges.

We randomly removed 4 out of 38 stations (ca. 10% of all stations) before clustering and repeated this five times. The clustering remained the same for all, except for 2 stations, which once or twice were assigned to Cluster 1 instead of Cluster 3. Next, we in a similar way also tested the effect of randomly removing 8 out of 38 stations (ca. 20%). The clustering remained the same for all, except for one station. Based on this, we conclude that the clustering is robust and not very sensitive to the removal of individual gauges.

In the manuscript we added the following text to the method's Section 2.3.2: "*A sensitivity analysis showed that this clustering approach is robust and that removing single or multiple catchments does hardly influence the clustering results. Furthermore,* a comparison with clusters derived by k-means clustering showed that the final clusters identified are relatively stable independent of the choice of the clustering technique." (L216)

L 294ff: I am not convinced here. If the cumulative sSSY is above average, this CAN mean that sediment stores are being depleted – OR that there has been a mass movement event and availability is greater than usual/before, which would imply greater instead of limited sed availability for coming events, right? The same goes for L 388 f.

**Reply**: Yes, however, the proxy we have used for sediment availability does not account for the actual (physical) causes of sediment availability. Instead, it compares the long-term cumulative sSSY with the yield expected based on the 10-12 years of data for that catchment. Nevertheless, the effects of smaller or larger mass movement events occurring on a more regular basis and the sediment transport down the river are represented by the mean annual sSSY curve. However, this method does not account for rare and unseen events that strongly affect the sediment availability in the catchment.

In your example, where the cumulative sSSY is above average, the sediment transport by the river has been higher than expected based on the 10-year mean. Even if a mass movement was the cause of this increase in sediment transport by the river, it is still unlikely that a next event will result in even more sediment transport based on the mean annual cumulative sSSY curve.

To better explain how this proxy has been created, we added the following figure to the Supporting Information:

[Figure]

Figure 1. *Proposed new figure to add to the Supporting Information. This figure explains the proxy for sediment memory by showing how we compare the long-term cumulative sSSY with the catchment specific mean annual cumulative sSSY curve.*

L403: "suggesting the presence of abundant sed sources" - or rather suggesting that sediment availability is not limiting?

**Reply**: We adjusted the sentence as suggested to:

"[…], *suggesting that sediment availability is not a limiting factor for riverine sediment transport.*" (L396)

L495ff: I find this comparison of events to the mean soil loss rate difficult, because the latter can vary greatly between catchments – as you state yourself later. Not sure it is needed here?

**Reply**: We removed this comparison and instead focus more on the sSSY event fraction in relation to the catchment mean annual sSSY (which we address in the next paragraph). Based on this comment (and an earlier comment) we made significant modifications to the 3rd and 4th paragraphs of section 4.2. Please see the manuscript with marked changes for the new text (L490-508).

L 522ff: I agree with this model assumption, and this is what Schmidt et al (2022) showed, that once areas above 2500 m become snow free glaciers also start to become snow-free and start to melt. Also compare to hydrograph separation studies such as Kormann et al. 2016, expecially figure 3 (10.3390/hydrology3010007).

**Reply**: Thank you for highlighting these studies. We address this comment by making the link to the work by Schmidt et al. and Kormann et al.:

"*Furthermore, we also noticed that glacial melt occurs only when the snow cover has disappeared or is at least smaller than 30%. This could be a model artefact, as it is consistent with how glacial melt is represented in the PCR-GLOBWB 2.0 model, which assumes that glaciers only start melting when they are no longer covered (and protected) by snow. This assumption is supported by findings of Schmidt et al (2022), who showed that subglacial*

*sediment sources are inactive as long as areas above 2500 m (including glacier tongues) are frozen or snow-covered, and of Kormann et al. (2016)."* (L520)

L 599: This is a strong assumption. Have you tested this by applying it to a catchment that is not in your dataset? How well does it work there? If not, I think it would be good to test this.

**Reply**: We address this comment by rephrasing this part so that we focus more on the generalizability of the methods rather than the results. Nevertheless, we believe that our results could also be generalised, but only under the condition that the unseen catchments have similar characteristics to the catchments considered in our dataset. We clarified this in the text as follows:

"Our choice of methods and the use of a large sample of catchments contribute to the generalisability of our methods and results. *The clustering of SSC regimes and the event classification scheme are designed to be transferable to other catchments and regions. Thus,* this study does not only improve our understanding of the complex hydrological-sedimentary response in the studied catchments, but *it also opens the door to further large-sample studies focusing on SSC. Finally, our results and conclusions might be generalizable to other mountain regions with similar characteristics, because the catchments considered in our dataset cover a variety of catchment characteristics (mean catchment elevation from 590m to 2889m, glacier cover from 0% to 33%, fraction of precipitation falling as snow from 0 to 0.34)."* (L595)

Minor comments:

L2: detrimental impact on water use – you mean water quality?

**Reply**: Yes, we mean the impact on water quality which will influence water use. We replaced "water use" by "water quality" to clarify this. (L2)

L11: start a new paragraph as you move to events?

**Reply**: Thank you for this suggestion, which we have adopted. (L7)

L13: "based on their dominant transport processes" – this raises questions rather than explaining things? I suggest to either elaborate or explain what you mean by dominant transport processes and how you get there, or remove this bit.

**Reply**: We removed this to avoid confusion.

L 22f: "Our findings underscore the importance of […] extreme SSC events on water quality…" – do they? Isn't it rather that you present a new systematic way to analyze? Consider whether something along the lines of the sentence in L 87f. would be more appropriate here (improves understanding of sed dynamics not just at the local scale and enables generilazation across catchments).

**Reply**: Based on your first comment, we rewrote our Abstract and this sentence did not make it to our updated and final version.

L 95: what do you mean by "external"? I suggest to remove this, it kind of doubles with "between catchments" and is not necessary. This repeats later (L172), maybe in other places as well.

**Reply**: Thanks for this suggestion. We removed the word "external" as it is indeed unnecessary.

L 123: "reconstruct"? Maybe find a different word (reconstruct sounds like it has been there before, when really you use SSC from samples and turbidity measurements to compute continuous SSC…)

**Reply**: We replaced "reconstruct" with "derive". (L107)

L124: this is a small thing, but technically, you use the short form NTU before the explanation in the line below.

**Reply**: We introduced the explanation of the short form of NTU when it is first mentioned: "[…] turbidity levels > 1000 *nephelometric turbidity units (*NTU*)*) […]" (L108)

L147: "on a finer time scale" – at higher temporal resolution?

**Reply**: We replaced "on a finer time scale" with "at higher temporal resolution". (L131)

L 263: on average 73 per catchment – can you add the min and max as well? This would help to understand the range better.

**Reply**: We added the range (minimum and maximum number of events per catchments) to the text: "[…] events per catchment (min = 10, max = 143)." (L251)

L298f: "we divided the deviation…." – this is really difficult to understand. Can you add a formula or graph to illustrate this?

**Reply**: We clarified this in the text by changing the sentence to: "To make this index comparable among catchments, we *standardized* the deviation of the long-term cumulative sSSY from the catchment specific mean annual sSSY curve *by dividing it by* the catchment annual mean cumulative sSSY". (L288)

In addition, we added an additional figure to the Supporting Information with a more detailed visualisation (including figures and formula's) of how we obtained this proxy (see also earlier comment and Figure 1).

L308: you mean sediment output or export not input, right? Also, Schmidt et al. have worked in these catchments as well. They also state that the glacier cover is a little lower, according to the glacier inventory of 2015. Did you calculate the glacier cover yourself?

**Reply**: I replaced "input" with "output" (L299). Furthermore, we did not use the glacier inventory of 2015, but we used the fraction of glaciers as provided by LamaH-CE and CAMELS-CH (see Figure S3 in the Supporting Information). Both datasets used the land classes from the CORINE Land Cover 2012 raster dataset to derive glacier cover. We decided to use these datasets because they are available for Switzerland and Austria, are commonly used, and apply the same methods to extract catchment characteristics, which is important for comparability among catchments (and countries).

L361: "glaciers have become snow-free" – this sounds like they have to become entirely snow-free. I think it is better so say that snow cover is minimal, because it may remain in the upper parts.

**Reply**: We rephrased to: "[…] *when the snow cover on glaciers is minimal and the glacier becomes susceptible to melting.*" (L352)

L 372f: "when we look at the average of all events per type (ignoring the outliers)…" – sorry, I don't understand this, maybe it's not just me and you could rephrase this?

**Reply**: We removed *"(ignoring the outliers)"* because it is confusing. We want the reader to focus on the median of all events per type (as visualised in the boxplots in Figure 6j). Therefore, we adjusted the text to:

"[…] when we look at the *median* of all events per event type, other event types result in higher peak SSC *(Figure 6j)* or *peak* sSSY (Figure S8j in Supporting Information)." (L364)

L384f: Events partly driven by glacier melt occur once snow cover is smaller than 30%: obviously, because there is no glacier melt beforehand, which only starts once the snow cover on the glaciers is removed. Compare to Schmidt et al.

**Reply**: In the results section, we prefer to only describe the observations without any further interpretation. In the discussion section, we do discuss these results in more detail, also in the light of the results by Schmidt et al. (see sections 4.1 and 4.2).

Fig. 7: some of the bars are so transparent that it is very hard to see them – at least on my device. You might want to improve this?

**Reply**: We addressed this comment by setting the colour bar to the log scale. This has improved the readability of Figure 7.

L 409: "three clusters […] which are similar in terms of static attributes" – wait, the clusters are not similar right? But the stations within a clusters? This is unclear to me, consider rephrasing.

**Reply**: We rephrased this to: "three clusters […] *where the catchments in each cluster share similar static attributes (Figure 4a-h) and seasonal regimes of snow- and glacial melt (Figure 3d-i).*" (L402)

L414: The sentence starting here is very long (until L 417). Consider creating several sentences here.

**Reply**: We rewrote this section by subdividing it into two separate sentences: *"The importance of snow- and glacial melt is confirmed by the analysis of the static catchment characteristics (Figure 4a-h). To explain the seasonal sediment dynamics, the most important characteristics are those that influence the magnitude and timing of snow- and glacial melt, for example, the presence of glaciers (even if they cover only a small part of the catchment), the catchment elevation, the proportion of precipitation falling as snow, and the catchment area."* (L407)

L418ff: I think there are many more studies showing this (e.g. by Delaney et al., Schmidt et al., Hinderer et al., ….?). I suggest to either cite more here or say "for example" or "just to name a few" or so.

**Reply**: We rephrased this to: "The importance of meltwater for SSC has already been highlighted by *multiple* other studies*, for example*: […]*"* (L411)

L422ff: so sediment stored in recently deglaciated moraines is then transported in subglacial channels? I think this needs to be restructured and clarified..

**Reply**: We restructured and clarified this part as follows: "The high concentration of suspended sediment in glacial meltwater can be explained by the importance of the glacier and glacier forefields as a sediment source and temporary storage. *Glacial abrasion on the bedrock causes*

*erosion and these sediments get transported by subglacial channels. However, these channels need time to develop at the start of the ablation season before they become very effective (Swift et al.,2005). In addition, a lot of material is stored in the recently deglaciated moraines and the glacial forefields (Moore et al., 2009). The braided stream network in this deglaciated terrain can be highly dynamic and therewith has an important control on the sediment availability (Mancini et al., 2024). This means that* areas above 2500 m.a.s.l., which are characterized by glacier tongues, bare rock surfaces and recently deglaciated areas, are crucial for sediment generation in high-altitude catchments (Schmidt et al.,2022). A sudden release of suspended sediments has been observed when areas above 2500 m become snow-free and the sediments are no longer protected from mobilization by the snow cover (Schmidt et al., 2022)." (L415-424)

L 435: Schmidt et al (2023) have also shown that sediment rating curves do not work well here.

**Reply**: We added Schmidt et al. (2023) as an additional reference here, as they show that a multivariate QRF approach performs favourably compared to rating curves. (L430)

L 441: Have you considered Zhang et al. (2021) as well here? (10.1029/2021WR030690)

**Reply**: We think the work of Zhang et al. is indeed worth mentioning here and have added it as an example: "One of the major limitations is that the Q-SSC relationship does not explicitly address the sources of sediment and their activation by different hydro-climatic forcing. Costa et al. (2017) have shown that a sediment source perspective using hydro-climatic forcing (rainfall, snowmelt and glacial melt) is more appropriate to explore sediment dynamics than traditional discharge-based rating curves. *In line with this, Zhang et al. (2021) introduced a sediment-availability-transport model that captures the time-varying sediment availability.* Their results are consistent with ours, as our results suggest that the origin of water entering the river system is more informative for understanding the mean annual SSC regime than the absolute magnitude of discharge alone." (L433)

L485: Schmidt et al 2022 use the same gauge, so it is not "in other alpine catchments"?

**Reply**: Yes, Schmidt et al. used the same gauge as we did, while Mano et al. analysed the SSC in four alpine catchments in France, which we did not include in our study. We rephrased to: "[…] concentrations in the same order of magnitude have been found *by other authors (Schmidt et al., 2022) and in other alpine catchments (Mano et al., 2009)."* (L483)

L512: you mean up to 280% of the AVERAGE annual sediment yield here, right?

**Reply**: Yes indeed. Thanks for the correction. We added the missing word "average". (L504)

L 525: … Sediment concentrationS may not rise it THE sediment input is proportional to THE water input"?

**Reply**: Corrected to: "*[…]* sediment *concentrations* may *not increase due to dilution effects* (Lana-Renault et al., 2007) if *the* sediment input is proportional to *the* water input." (L514)

L 534: I think you mean that you reset the memory manually in your calculations, correct? First I thought there was some process happening in the catchments, so I think it is important to phrase this in the active instead of passive here.

**Reply**: Rephrased to: "[…] no relationship was visible when *we reset the catchment memory* each winter, implying that sediment stores build up and empty over several years." (L530)

L 553: particle size, shape… and color?? (Merten et al., 2014, 10.1007/s11368-013-0813-0)

**Reply**: In addition to particle size, shape and density, color is indeed an additional important characteristic that influences the relation of SSC with turbidity. We extended the list between brackets and added Merten et al. as a reference: "[…] the varying characteristics of the sediments in suspension (particle size, shape, density, *and color*) (Gippel, 1995*; Merten et al., 2014) […]*" (L550)

L 566: so k-means does not have the same limitations?

**Reply**: Rephrased to: "*As each clustering method has its advantages and disadvantages, we performed a sensitivity analysis comparing the results of hierarchical clustering with k-means clustering. This showed that the clusters identified are relatively stable, regardless of the choice of the clustering technique. However, a well-known disadvantage of clustering in general is that the selected indicators can strongly influence the final clusters. We deliberately selected indicators that were able to capture […]*" (L561)

L 620: "shorter periods with reduced snow cover" – I think you mean shorter periods during which there is snow cover and generally reduced snow cover?

**Reply**: Rephrased to: "Consequently, *shorter periods with snow and generally reduced snow cover* will lead to […]" (L620)

L621: increased connectivity or rather longer periods where connectivity is given or can be established?

**Reply**: Rephrased to: "[…] will lead to *longer periods with potentially* increased connectivity of sediment sources to the river, […]" (L621)

L 624: projected and future is kind of the same.

**Reply**: We removed the redundancy: "[…] the impact of *future* precipitation […]" (L624)

L 647: "… the total annual sed yield of a catchment" – I am assuming this is an "average catchment"? This is debatable due to the large differences between catchments..

**Reply**: We corrected this to: "the total annual sediment yield of an *average* catchment" (L647)

Li et al., *Air Temperature Regulates Erodible Landscape, Water, and Sediment Fluxes in the Permafrost-Dominated Catchment on the Tibetan Plateau, Water Resources Research, 57, doi: 10.1029/2020WR028193, 2021.*

Schmidt et al., *Suspended sediment and discharge dynamics in a glaciated alpine environment: identifying crucial areas and time periods on several spatial and temporal scales in the Ötztal, Austria, Earth Surface Dynamics, 10, 3, 653-669, doi: 10.5194/esurf-10-653-2022, 2022.*

Kormann et al., *Model-Based Attribution of High-Resolution Streamflow Trends in Two Alpine Basins of Western Austria, Hydrology, 3, 1, 7, doi: 10.3390/hydrology3010007, 2016.*

Swift et al., *Basal sediment evacuation by subglacial meltwater: suspended sediment transport from Haut Glacier d'Arolla, Switzerland, Earth Surface Processes and Landforms, 30, 7, 867-883, doi: 10.1002/esp.1197, 2005.*

Moore et al., *Glacier change in western North America: influences on hydrology, geomorphic hazards and water quality, Hydrological Processes, 23, 1, 42-61, doi: 10.1002/hyp.7162, 2009.*

Mancini et al., *Rates of Evacuation of Bedload Sediment From an Alpine Glacier Control Proglacial Stream Morphodynamics, Journal of Geophysical Research: Earth Surface, 129, 8, doi: 10.1029/2024JF007727, 2024.*

Zhang et al., *Constraining Dynamic Sediment-Discharge Relationships in Cold Environments: The Sediment-Availability-Transport (SAT) Model, Water Resources Research, 57, 10, doi: 10.1029/2021WR030690, 2021.*

Costa et al., *A Process–Based Rating Curve to model suspended sediment concentration in Alpine environments, Hydrology and Earth System Sciences Discussions, 1-23, doi: 10.5194/hess-2017-419, 2017.*

Mano et al., *Assessment of suspended sediment transport in four alpine watersheds (France): influence of the climatic regime, Hydrological Processes, 23, 5, 777-792, doi: 10.1002/hyp.7178, 2009.*

Gippel, *Potential of turbidity monitoring for measuring the transport of suspended solids in streams, Hydrological Processes, 9, 1, 83-97, doi: 10.1002/hyp.3360090108, 1995.*

Merten et al., *Effects of suspended sediment concentration and grain size on three optical turbidity sensors, Journal of Soils and Sediments, 14, 7, 1235-1241, doi: 10.1007/s11368-013-0813-0, 2014.*